# Optimizing Rank for High-Fidelity Implicit Neural Representations

Julian McGinnis [1 2 3 4]   Florian A. Hölzl [2 5]   Suprosanna Shit [6]   Florentin Bieder [7]   Paul Friedrich [7]
Mark Mühlau [3]   Bjoern Menze [6]   Daniel Rueckert[† 2 4]   Benedikt Wiestler[† 1 4]

## Abstract

Implicit Neural Representations (INRs) based on vanilla Multi-Layer Perceptrons (MLPs) are widely believed to be incapable of representing high-frequency content. This has directed research efforts towards architectural interventions, such as coordinate embeddings or specialized activation functions, to represent high-frequency signals. In this paper, we challenge the notion that the low-frequency bias of vanilla MLPs is an intrinsic, architectural limitation to learn high-frequency content, but instead a symptom of stable rank degradation during training. We empirically demonstrate that regulating the network's rank during training substantially improves the fidelity of the learned signal, rendering even simple MLP architectures expressive. Extensive experiments show that using optimizers like Muon, with high-rank, near-orthogonal updates, consistently enhances INR architectures even beyond simple ReLU MLPs. These substantial improvements hold across a diverse range of domains, including natural and medical images and novel view synthesis, with up to +9 dB PSNR over the same architecture. Code is available here.

## 1. Introduction

INRs, popularized by their usage in NeRFs (Mildenhall et al., 2021), have emerged as a versatile framework for modeling diverse signal modalities including shapes (Park et al., 2019; Gropp et al., 2020; Davies et al., 2020; Sitzmann et al., 2020a), images (Tancik et al., 2020; Sitzmann et al., 2020b; Mehta et al., 2021), scenes (Mildenhall et al., 2021; Barron et al., 2021), videos (Chen et al., 2021; 2022) and medical data (Wolterink et al., 2022; McGinnis et al., 2023; Friedrich et al., 2025). The vast progress of this field has been enabled by architectural improvements to the vanilla ReLU MLP, ranging from coordinate encodings (Tancik et al., 2020; Müller et al., 2022) to non-conventional activations (Sitzmann et al., 2020b; Ramasinghe & Lucey, 2022) that address the spectral bias of vanilla ReLU networks (Rahaman et al., 2019; Basri et al., 2020) and enable the representation of fine details. Yet, this pursuit of expressiveness has created a dilemma. Architectures with non-linear activations like SIREN (Sitzmann et al., 2020b) and WIRE (Saragadam et al., 2023), while powerful, often lack implicit regularization (Ramasinghe et al., 2022) for robust performance on ill-posed inverse problems, where overly expressive architectures may lead to oscillatory patterns (Ramasinghe et al., 2022; Kim & Fridovich-Keil, 2025) and require carefully chosen regularization to enable smooth interpolation behavior (Liu et al., 2022; Niemeyer et al., 2022). At the same time, a neural network with a low-frequency and thus a smooth inductive bias, such as a vanilla ReLU network (Rahaman et al., 2019), may be an ideal candidate for inverse problems but may fall short due to its limited expressiveness in fitting high-frequency data.

In this paper, we present a new perspective on this dilemma by shifting focus from architecture to optimization. While prior work has primarily addressed limited expressiveness through architectural modifications (Sitzmann et al., 2020b), we take a different approach. Rather than analyzing INRs through the conventional lenses of Fourier analysis (Yüce et al., 2022) or the Neural Tangent Kernel (NTK) (Jacot et al., 2018), we take a *stable rank perspective* (Daneshmand et al., 2020; Feng et al., 2022; Ramasinghe & Lucey, 2022). This viewpoint offers novel insights beyond architectural considerations into the dynamics of network weights during training, enabling us to address fundamental problems in current INR architectures through principled optimization strategies. We make the following contributions:

---

[†] Equal senior contribution. [1]Chair for AI for Image-Guided Diagnosis and Therapy, Technical University of Munich, Germany [2]Chair for AI in Medicine, Technical University of Munich, Germany [3]Department of Neurology, Klinikum Rechts der Isar, Technical University of Munich, Germany [4]Munich Center for Machine Learning (MCML), Germany [5]Hasso-Plattner-Institute, University of Potsdam, Germany [6]Department of Biomedical Engineering, University of Zurich, Switzerland [7]Center for medical Image Analysis and Navigation (CIAN), University of Basel, Switzerland. Correspondence to: Julian McGinnis <julian.mcginnis@tum.de>.

*Proceedings of the $43^{rd}$ International Conference on Machine Learning*, Seoul, South Korea. PMLR 306, 2026. Copyright 2026 by the author(s).

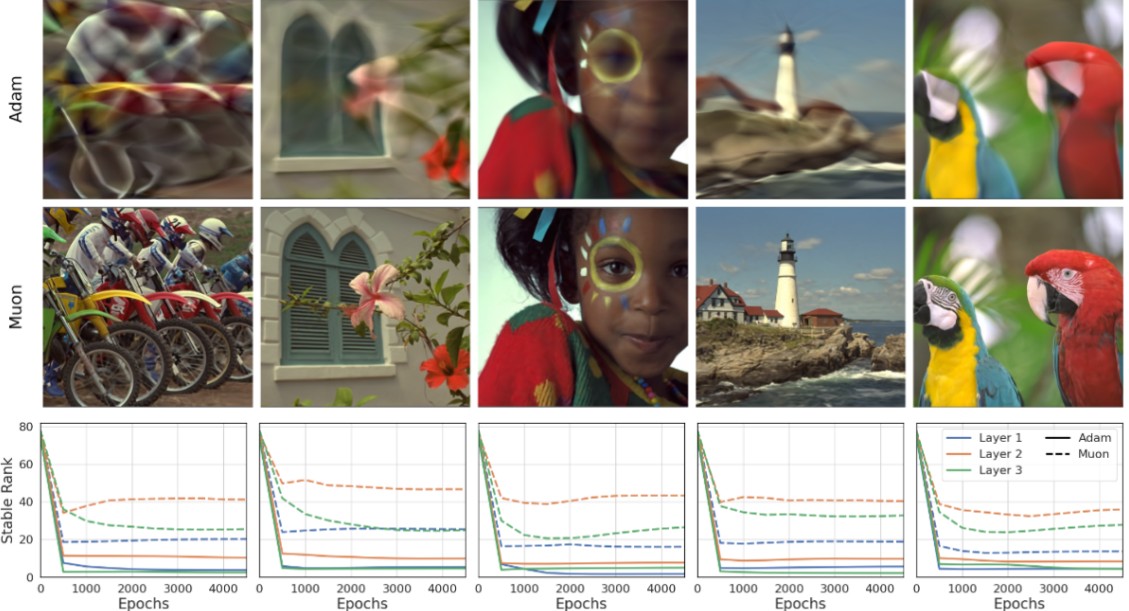

*Figure 1. Rank preservation overcomes the low-frequency bias* of ReLU MLPs. Vanilla ReLU MLPs lack high-frequency detail (top). Orthogonalized, rank-preserving weight updates enable them to achieve faithful high-frequency reconstructions (middle) by maintaining a stable layer rank throughout optimization (illustrated for hidden layers trained with Adam, which exhibits rank collapse, versus Muon, which preserves stable rank; bottom).

- We propose the stable rank as a key measure of expressiveness in INRs, and argue that the inability of vanilla MLPs to fit high-frequency signals is due to rank degradation during training.
- Based on this insight, we provide a unifying framework that explains the effectiveness of common architectural modifications in INRs, such as coordinate embeddings and alternative activation functions, in learning high-fidelity implicit functions.
- To explicitly address rank degradation, we shift focus from architecture to optimization and propose Muon, an orthogonalizing optimizer, as a natural remedy.
- Extensive experiments demonstrate its effectiveness, significantly improving performance for all current architectures across various modalities and applications.

## 2. Related Work

The field of implicit representations has expanded significantly from its origins in shapes (Park et al., 2019) and neural radiance fields (Mildenhall et al., 2021), largely due to architectural improvements that have improved the expressiveness of INRs. Recent work has presented further architectural modifications (Chen et al., 2023a;c; Kazerouni et al., 2024; Xie et al., 2023; Liu et al., 2024; Cai et al., 2024a;b), proposed meta-learning strategies for faster convergence (Tancik et al., 2021; Sitzmann et al., 2020a; Tack et al., 2023), improved initialization schemes (Saratchan-

dran et al., 2024; Kania et al., 2024; Koneputugodage et al., 2025; Yeom et al., 2024) and introduced prioritized coordinate sampling schemes (Kheradmand et al., 2024; Zhang et al., 2024a;b; 2025). Adjacent fields have explored conditional network architectures (Park et al., 2019; Mehta et al., 2021; Dupont et al., 2022b), enabling the efficient modeling of signal datasets and compressing them into latent representations (Dupont et al., 2022b; You et al., 2023; Friedrich et al., 2025). Dupont et al. (2022a) have identified INRs as a new data modality, fueling research in scaling INRs from individual instances to datasets (Ma et al., 2024; Papa et al., 2024), and proposing novel methodologies for learning in permutation-invariant weight spaces (Zhou et al., 2023; Navon et al., 2023). Novel architectures for INRs have been primarily investigated through the lens of the Neural Tangent Kernel (Jacot et al., 2018; Tancik et al., 2020; Liu et al., 2024; Cai et al., 2024a) and Fourier analysis (Yüce et al., 2022; Ramasinghe et al., 2022; Lindell et al., 2022). While these remain insightful and comprehensively explain the success of recent architectural modifications (Tancik et al., 2020; Cai et al., 2024a), we believe that providing a common framework with respect to the architecture *and* optimization dynamics using the stable rank allows us to unify previous approaches in a common framework.

With the notable exception of Saratchandran et al. (2023); Chng et al. (2025), the INR community has not yet questioned the selection of alternative optimizers over Adam (Kingma & Ba, 2014). The vast majority of both

early and contemporary INR applications and studies continue to rely on common first-order optimizers (Tancik et al., 2020; Sitzmann et al., 2020b; Mildenhall et al., 2021; Ramasinghe & Lucey, 2022; Saragadam et al., 2023; Liu et al., 2024; Essakine et al., 2025; Kim & Fridovich-Keil, 2025). Interestingly, in the broader domain of neural fields, particularly within physics-informed neural networks (PINNs) (Xie et al., 2022), curvature-aware optimizers such as L-BFGS (Liu & Nocedal, 1989) are frequently employed, often following an initial phase of Adam optimization (Rathore et al., 2024). Such hybrid optimization is particularly effective for PINNs, whose physics-based residuals often yield sharp and highly non-convex loss landscapes. Within the context of INRs, L-BFGS tends to produce well-conditioned gradients for sinusoidal or Gaussian activations, but remains prohibitively expensive and scales poorly with model size (Saratchandran et al., 2023).

Chng et al. (2025) offer a complementary perspective by studying curvature-aware preconditioners (Dauphin et al., 2015; Yao et al., 2021) for accelerated convergence relative to Adam, while reaching comparable reconstruction quality. They further note their ineffectiveness for ReLU MLPs, explaining Adam's continued prevalence in INRs.

Taken together, prior work has pushed INRs forward almost exclusively through architectural innovations, while optimization has remained an implicit afterthought despite clear evidence from adjacent domains that optimization can fundamentally alter model behavior. What is missing is a measure that can jointly reflect architectural bias and the dynamics induced by different optimizers. We argue that *stable rank* serves precisely this role. While it remains sparsely examined in INR literature (Ramasinghe & Lucey, 2022), it has become a central tool in deep learning for characterizing generalization and optimization dynamics (Daneshmand et al., 2020; Martin & Mahoney, 2021; Noci et al., 2022; Feng et al., 2022; Sanyal et al., 2019; Geshkovski et al., 2025). The stable rank captures how architecture and optimization interact during training and connects them to model expressiveness (Dong et al., 2021; Boix-Adsera et al., 2023; He et al., 2023; Geshkovski et al., 2025; Wu et al., 2024). This makes stable rank a natural quantity for unifying architectural interventions of INRs with our new perspective on optimization to learn higher-frequency functions.

## 3. A Stable Rank Perspective on Spectral Bias

The phenomenon of spectral bias (Rahaman et al., 2019) describes how neural networks prioritize low-frequency functions during training. Rather than analyzing this in the function's Fourier spectrum, we examine how it arises from the linear–algebraic structure linking activations and weights through gradients. Based on the stable rank of layer updates, we clarify how activations shape layer weights, why this

reinforces a low-frequency bias in INRs, and how different interventions address this bias.

### 3.1. Stable Rank and Model Expressiveness

An INR parameterizes a signal as a continuous function $f_\theta : \mathbb{R}^d \to \mathbb{R}^c$, where $\theta$ denotes the learnable parameters, $d$ is the dimensionality of the input coordinate space, and $c$ is the output dimension. For example, in the case of image representation, $f_\theta : \mathbb{R}^2 \to \mathbb{R}^3$ maps 2D spatial coordinates to RGB color values. The function $f_\theta$ is typically parameterized by an MLP, i.e., $\theta$ represents the weights and biases of the neural network.

**Activations Constrain Updates:** Consider a single layer $l \in \{0, \dots, L-1\}$ with a batch of $B$ input activations $H_l \in \mathbb{R}^{d_l \times B}$, weights $W_l \in \mathbb{R}^{d_{l+1} \times d_l}$, biases $b_l \in \mathbb{R}^{d_{l+1}}$, and pre-activation output $H_{l+1} = W_l H_l + b_l \mathbf{1}^\top \in \mathbb{R}^{d_{l+1} \times B}$, where $\mathbf{1} \in \mathbb{R}^B$ is the all-ones vector.

Let $G_{l+1} = \nabla_{H_{l+1}} \mathcal{L} \in \mathbb{R}^{d_{l+1} \times B}$ be the upstream gradient.

The gradient with respect to the weights is correspondingly defined as the outer product $\nabla_{W_l} \mathcal{L} = G_{l+1} H_l^\top \in \mathbb{R}^{d_{l+1} \times d_l}$. The bias gradient $\nabla_{b_l} \mathcal{L} = G_{l+1} \mathbf{1}$ lies in the column space of $G_{l+1}$ and does not alter the row-space inclusion below; we therefore omit it from the subsequent rank analysis. This factorization implies that the subspaces are related:

$$\text{rowspace}(\nabla_{W_l} \mathcal{L}) \subseteq \text{colspace}(H_l), \quad (1)$$

i.e., the row space of the update is contained in the span of the batch activations, meaning that learning is dependent on the dominant activation directions.

To quantify the effective dimensionality of this process, we use the *stable rank* $s(A)$ as a numerically stable measure of diversity for a matrix $A$ with singular values $\sigma_i$:

$$s(A) = \frac{\|A\|_F^2}{\|A\|_2^2} = \frac{\sum_i \sigma_i^2}{\sigma_{\max}^2}.$$

The stable rank measures how distributed the energy of a matrix is among its singular values. This allows us to formulate how the update is dependent on the effective dimensionality of the activations. A rank-1 matrix has $s(A) = 1$, while a semi-orthogonal matrix with rank $k$ has $s(A) = k$.

For $Y = AB$ with matrices $A \in \mathbb{R}^{m \times n}$ and $B \in \mathbb{R}^{n \times p}$, the stable rank of $Y$ satisfies $s(Y) \leq \text{rank}(Y) \leq \min\{\text{rank}(A), \text{rank}(B)\}$. Applying this inequality to the batched gradient $\nabla_{W_l} \mathcal{L} = G_{l+1} H_l^\top$ gives

$$s(\nabla_{W_l} \mathcal{L}) \leq \text{rank}(H_l). \quad (2)$$

Thus, for a single layer, the input activations upper-bound, but do not fully determine, the effective dimensionality of

*Figure 2.* Reconstructing a tiger from the animal AFHQ dataset using a vanilla ReLU MLP and different rank-preserving/inducing methods. We propose to use Muon, which induces a high stable rank via its orthogonalized weight updates.

the update. While the upstream gradient $G_{l+1}$ can in principle align with any direction, the rank of $\nabla_{W_l}\mathcal{L}$ cannot exceed that of $H_l$, so a degenerate activation subspace restricts the directions along which the weights can move.

**From Layer to Network:** To analyze a network's capability to fit high-frequency functions, we must examine the sequence of network layers, starting from the input. In coordinate-based networks such as INRs, the input batch $H_0 \in \mathbb{R}^{2 \times B}$ for raw 2D coordinates lies in a space of rank at most two, yielding $s(\nabla_{W_0}\mathcal{L}) \leq 2$. Through successive layers, the mappings $H_{l+1} = f_{l+1}(H_l)$ form a sequence of compositions that can only reduce, or at best preserve, the span of activations. Each layer thus acts on activations already restricted to a subspace determined by the preceding layer, leading to deeper layers operating on progressively lower-dimensional subspaces. While this compression is strict for networks with linear layers, the key property of nonlinearities is the possibility to locally increase rank. Without other adaptations, the properties of the nonlinearities are crucial to keep the overall network function expressive (Ramasinghe & Lucey, 2022). However, the monotonic decrease in network expressiveness with depth is general, and the speed or severity of degradation depends on interventions at the architecture- or optimization-level (Feng et al., 2022). This problem of rank reduction is empirically known, with e.g. Daneshmand et al. (2020) showing rank collapse in the last layer activation and weights, or Huh et al. (2021) showing that deeper networks struggle to fit high-rank linear functions. Stable rank allows one not only to quantify network fidelity through analyzing network activations and weights but also to target model performance through principled interventions based on the rank.

### 3.2. Interventions to Increase Stable Rank

Several techniques for training INRs can be reinterpreted as interventions designed to break the low stable rank bottleneck. We group these interventions into two broad categories of architectural modifications. First, adapting the input to the INR, and second, improving the signal propagation and thus the span of the activations within the network.

**Input-Level Interventions:** Methods such as Fourier Features (Tancik et al., 2020) map the low-dimensional in-

put coordinate $x \in \mathbb{R}^d$ to a high-dimensional vector $\gamma(x) \in \mathbb{R}^{2 \cdot d \cdot F}$ using sampled random frequencies $F$: $\gamma(x) = [\cos(2\pi Bx), \sin(2\pi Bx)]$, where $B \in \mathbb{R}^{F \times d}$ contains entries sampled i.i.d. from a normal distribution $\mathcal{N}(0, \sigma^2)$. This expands the batch matrix $H_0$ to $H' = \gamma(H_0)$. Under common sampling schemes (e.g., uniform on $[0,1]^d$), the columns of $\gamma(H_0)$ are approximately decorrelated across frequencies, so $\text{rank}(H'_0)$ scales with the number of frequency bands $F$ (until it saturates at $B$). This increases the stable rank bound on the first-layer update $s(\nabla_{W_1}\mathcal{L}) \leq \text{rank}(H'_0)$, enabling learning of higher-frequency components. The trade-off is that frequency bands are static and hand-engineered, may not match the data and/or task. For example, for Fourier Feature Networks (FFNs) (Tancik et al., 2020), the choice of $\sigma$ is critical and architecture/task-dependent; we examine the interaction between optimizer and architecture-specific hyperparameters empirically in Table 21 and Appendix D.3. Similarly, plane-based methods (Sivgin et al., 2025), analogous to input-level interventions, raise rank via learned feature grids. Yet, as we show in Appendix D.4, this alone does not prevent rank collapse in the decoder MLP.

**Activation-Level Interventions:** While input-level methods improve INR performance by indirectly increasing stable rank, they fail to overcome the intrinsic low-frequency bias. A direct approach targets the activations themselves to counter rank diminishing with depth. Normalization, particularly Batch Normalization (BN), has been shown to aid INR training for ReLU networks (Cai et al., 2024a;b), likely by enhancing the isotropy of $\text{colspace}(H_l)$ and thus raising the stable rank (Daneshmand et al., 2020) (*see Figure 2*). Alternatively, modifying activation functions can promote high-frequency learning. Sinusoidal Representation Networks (SIRENs) (Sitzmann et al., 2020b) use $\sin(\cdot)$ activations and specialized initialization to maintain gradient stability and represent high frequencies (Ramasinghe & Lucey, 2022), effectively increasing $\text{rank}(H_l)$ but at the cost of greater tuning complexity.

### 3.3. Optimizer-Level Interventions

While the previously introduced architectural interventions lead to substantial performance improvements of INRs, the sparse updates of SGD and Adam (see (Kingma & Ba,

*Table 1.* **Image overfitting performance**. Quantitative comparison of reconstruction quality for different architectures using the Adam optimizer versus the Muon optimizer. We report the mean and standard deviation for PSNR, SSIM, and LPIPS over 24 Kodak images. The best-performing optimizer for each architecture is highlighted in **bold**.

| Model | PSNR ($\uparrow$) | | SSIM ($\uparrow$) | | LPIPS ($\downarrow$) | |
|---|---|---|---|---|---|---|
| | Adam | Muon | Adam | Muon | Adam | Muon |
| ReLU MLP | $22.83_{\pm 2.85}$ | $\mathbf{30.04}_{\pm 3.59}$ | $0.546_{\pm 0.155}$ | $\mathbf{0.813}_{\pm 0.074}$ | $0.653_{\pm 0.142}$ | $\mathbf{0.290}_{\pm 0.094}$ |
| ReLU FFN | $31.21_{\pm 2.71}$ | $\mathbf{40.34}_{\pm 3.56}$ | $0.846_{\pm 0.041}$ | $\mathbf{0.966}_{\pm 0.012}$ | $0.192_{\pm 0.054}$ | $\mathbf{0.019}_{\pm 0.009}$ |
| Gauss MLP | $35.28_{\pm 1.85}$ | $\mathbf{40.81}_{\pm 2.75}$ | $0.929_{\pm 0.014}$ | $\mathbf{0.978}_{\pm 0.005}$ | $0.059_{\pm 0.020}$ | $\mathbf{0.013}_{\pm 0.008}$ |
| Gauss FFN | $25.20_{\pm 2.18}$ | $\mathbf{28.31}_{\pm 1.99}$ | $0.666_{\pm 0.156}$ | $\mathbf{0.782}_{\pm 0.136}$ | $0.359_{\pm 0.142}$ | $\mathbf{0.206}_{\pm 0.107}$ |
| SIREN | $39.33_{\pm 2.64}$ | $\mathbf{41.43}_{\pm 2.94}$ | $0.965_{\pm 0.015}$ | $\mathbf{0.977}_{\pm 0.006}$ | $0.019_{\pm 0.015}$ | $\mathbf{0.011}_{\pm 0.006}$ |
| WIRE | $32.43_{\pm 9.56}$ | $\mathbf{42.99}_{\pm 3.34}$ | $0.794_{\pm 0.279}$ | $\mathbf{0.986}_{\pm 0.005}$ | $0.224_{\pm 0.324}$ | $\mathbf{0.007}_{\pm 0.006}$ |
| FINER | $39.87_{\pm 2.04}$ | $\mathbf{42.59}_{\pm 2.74}$ | $0.969_{\pm 0.007}$ | $\mathbf{0.982}_{\pm 0.005}$ | $0.013_{\pm 0.006}$ | $\mathbf{0.005}_{\pm 0.003}$ |

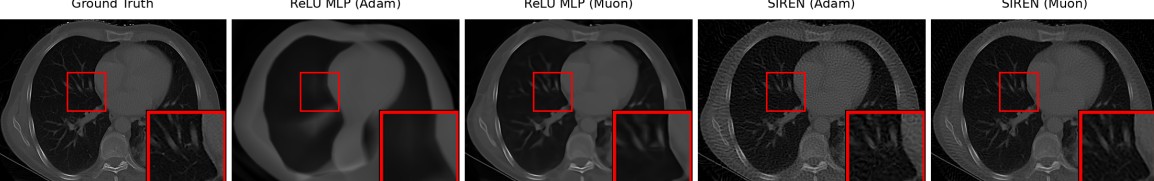

*Figure 3.* Qualitative comparison for CT image reconstruction using Adam and the orthogonalizing optimizer, Muon, for selected models.

2014)) lead to learning dynamics that work against the implicit interventions during optimization (Basri et al., 2020). Thus, we instead view this challenge as an optimization problem, calling for direct intervention at the update level. There has been little research in this direction for INRs, motivating us to apply rank-preserving pretraining, originally proposed for other domains (Daneshmand et al., 2020). By performing unsupervised gradient descent, minimizing the Frobenius norm $\|M_L\|_F^2$ of the activation covariance $M_L = H_L H_L^\top / N$, we can increase the stable rank of the final hidden layer. Optimizing this objective across mini-batches before the actual training leads to higher-rank activation patterns in the network, avoiding the directional gradient vanishing associated with rank degradation.

We quantify two rank-preserving interventions, *rank-inducing pre-training* and *batch normalization* (applied at different network stages), qualitatively in Figure 2 and quantitatively in Table 14 (App. C.1), and draw two observations. First, mitigating stable-rank decay improves reconstruction over the collapsing Adam baseline, confirming that rank preservation matters. Second, preserving rank alone is not sufficient: as anticipated for the activation-level interventions above, batch normalization keeps the stable rank highest of all, i.e., close to weight initialization values, yet improves reconstruction only marginally, because maintaining the stable rank near initialization impedes learning. Rank pre-training, by contrast, reaches a lower rank but does not interfere with the weight updates during training, and so reconstructs better while still avoiding rank collapse.

However, pre-training (even for multiple hidden layers) does not fully counteract the low-frequency bias in subsequent

training and thus only provides limited gains. Instead, we propose to explicitly enforce isotropy in the gradient updates to increase expressiveness. This can be achieved with the general class of orthogonalizing optimizers (Carlson et al., 2015; Tuddenham et al., 2022) that explicitly map each layer's update to a high-rank, near-orthogonal direction. Concretely, for an update matrix $U \in \mathbb{R}^{d_{l+1} \times d_l}$ with SVD $U = P\Sigma Q^\top$, we map $U$ to the Frobenius-nearest semi-orthogonal matrix by solving:

$$\text{Ortho}(U) = \underset{O^\top O = I}{\arg\min} \|O - U\|_F = PQ^\top. \quad (3)$$

This explicitly transforms an update $U$ with low stable-rank update into an update $\text{Ortho}(U)$ with *high stable rank*, where $s(\text{Ortho}(U)) = \text{rank}(\text{Ortho}(U)) = \min\{d_l, d_{l+1}\}$. In addition, the update is scale invariant ($\text{Ortho}(cU) = \text{Ortho}(U)$ for any $c > 0$). This forces the weight parameter update to be distributed evenly across all of its singular directions.

Experimentally, such maximal-stable-rank updates increase the rank of corresponding weight matrices compared to Adam, as shown in Figure 1 on single-image overfitting. In practice, we apply orthogonalizing updates to hidden-layer weight matrices and retain Adam for biases, input encodings, and other non-hidden parameters. This approach offers two core benefits over input and model-level interventions: (1) Generality, the optimization being model-agnostic and applicable to any 2D weight matrix, and (2) Adaptivity, operating on the actual per-step gradients to correct rank collapse as it emerges.

Moreover, working at the optimizer level naturally com-

poses with other interventions: input-level methods such as Fourier features or feature grids provide high-rank potential at the input, while the optimizer ensures the gradient dynamics do not collapse it. We confirm this for grid-based GA-Planes (Sivgin et al., 2025) in Appendix D.4, where orthogonalized updates restore the decoder's stable rank on top of the already structured input, yielding the best reconstruction quality.

A recent and low-overhead orthogonalizing optimizer is Muon (Jordan et al., 2024). It approximates the orthogonalization from Equation 3 with Newton–Schulz (Bernstein & Newhouse, 2024) instead of SVD, reducing the required matrix multiplications and making it efficient. We will thus focus on Muon in the following.

# 4. Experiments

To assess the impact of optimization with high-rank, near-orthogonal updates, we study two complementary tasks: (1) signal representation and (2) inverse problems. The first provides a controlled setting to analyze how optimization affects the learning of high-frequency details in signals such as images or shapes, which has become increasingly relevant for applications like image compression and shape representation (Dupont et al., 2021; Strümpler et al., 2022; Davies et al., 2020). The second focuses on ill-posed problems, such as single-image super-resolution, CT reconstruction, and novel view synthesis. This allows us to examine how orthogonal optimization with Muon (Jordan et al., 2024) interacts with architectural interventions and implicit regularization in comparison to conventional optimization with Adam. All experiments, with full optimizer and hyperparameter details, are provided in Appendix A and B.

## 4.1. Signal Representation

### 4.1.1. IMAGE OVERFITTING

Following Dupont et al. (2021); Saragadam et al. (2023), we evaluate image representation performance on the Kodak dataset (Kodak, 1991) ($768 \times 512$ resolution). Our setup exactly matches Saragadam et al. (2023), with all hyperparameters detailed in Appendix B.1. Results are reported in Table 1, with representative reconstructions shown in Figure 1. Orthogonalized optimization yields substantial and consistent improvements across all models. For ReLU-based INRs, the PSNR increases by $\sim$7–8 dB with Muon, bringing a plain **ReLU MLP to parity with FFNs** (Tancik et al., 2020) trained using the Adam optimizer. This is a remarkable result: despite the long-standing belief that standard MLPs cannot capture high-frequency content, orthogonalization of the updates alone closes this performance gap. Importantly, this holds across different $\sigma$ values for FFNs as demonstrated in Table 21, where Muon consistently

outperforms Adam-based FFNs. The optimizer-level improvement is thus additive on top of input-level interventions instead of a compensation for a poorly chosen $\sigma$, consistent with our framework's view that input- and optimization-level interventions act on distinct, complementary axes. Beyond ReLU networks, we also observe significant gains ($\sim$2–4 dB) for Gaussian-MLPs, SIREN, and FINER, highlighting that orthogonalized optimization improves reconstruction accuracy of INRs across different architectures.[1] These improvements not only hold for intensity-based PSNR and SSIM but also apply to the arguably more relevant perceptual metric LPIPS. As a further baseline, we compare against the sign-based optimizer Lion (Chen et al., 2023b) (Tables 18 and 19). Lion does not consistently preserve stable rank and trails Muon in both rank and reconstruction quality, indicating the gains come from orthogonalized updates rather than simply switching away from Adam.

### 4.1.2. AUDIO OVERFITTING

The frequency spectrum of audio varies substantially from spatially correlated image data. To assess 1D signal representation, we replicate the audio fitting task introduced by Sitzmann et al. (2020b) and later adopted in Essakine et al. (2025). We benchmark all methods on two signals: (1) a 6-second clip of *Bach's Cello Suite No.1: Prelude* and (2) a spoken sequence of digits (0–9). Our experimental setup exactly follows Sitzmann et al. (2020b), with all hyperparameter details and sweep results provided in Appendix B.1. Results in the main paper are shown for *Bach* in Table 2, while results for the spoken sequence are in Appendix C.3.

INRs optimized with Muon achieve substantially higher signal fidelity across all architectures, e.g., +12.5 dB SNR for a FFN and +9.6 dB for FINER, demonstrating that orthogonalized optimization generalizes effectively beyond images to 1D temporal signals. Notably, the ReLU MLP fails to reconstruct the full-length audio, independent of the optimizer. This is likely due to insufficient network capacity for representing long-duration waveforms. Evaluated on a shortened 200 ms segment (Appendix D.2), Muon-optimized ReLU MLPs successfully learn to reproduce the signal, confirming that the failure stems from capacity limitations rather than optimization instability.

### 4.1.3. SIGNED DISTANCE FIELD OVERFITTING

Next, we address the established problem of learning signed distance fields (SDFs) (Park et al., 2019; Liu et al., 2022; Coiffier & Béthune, 2024). Unlike images and audio, SDFs introduce global geometric and gradient consistency, offering a more structured and demanding setting that reveals how well an INR captures coherent 3D geometry. This task

---

[1] Real-valued variant of WIRE used. Complex-valued version performed inferior for both Adam and Muon in our experiments.

*Table 2.* **Audio overfitting performance (bach.wav).** Mean $\pm$ std over 5 seeds. Best-performing optimizer per row in **bold**.

| Model | SNR ($\uparrow$) | | SI-SNR ($\uparrow$) | |
|---|---|---|---|---|
| | Adam | Muon | Adam | Muon |
| ReLU MLP | 0.02 $_{\pm .00}$ | **0.14** $_{\pm .01}$ | −23.52 $_{\pm .39}$ | **−14.88** $_{\pm .47}$ |
| ReLU FFN | 8.88 $_{\pm .40}$ | **21.43** $_{\pm 1.11}$ | 8.31 $_{\pm .45}$ | **21.40** $_{\pm 1.11}$ |
| Gauss MLP | 8.76 $_{\pm .63}$ | **12.85** $_{\pm .32}$ | 8.16 $_{\pm .73}$ | **12.64** $_{\pm .34}$ |
| Gauss FFN | 37.80 $_{\pm .44}$ | **46.80** $_{\pm .91}$ | 37.80 $_{\pm .44}$ | **46.80** $_{\pm .91}$ |
| SIREN | 37.92 $_{\pm .19}$ | **47.46** $_{\pm .68}$ | 37.92 $_{\pm .19}$ | **47.46** $_{\pm .68}$ |
| WIRE | 3.65 $_{\pm .82}$ | **15.52** $_{\pm .60}$ | 1.23 $_{\pm 1.39}$ | **15.62** $_{\pm .63}$ |
| FINER | 27.22 $_{\pm .65}$ | **36.48** $_{\pm .81}$ | 27.21 $_{\pm .65}$ | **36.48** $_{\pm .81}$ |

*Table 3.* **SDF overfitting performance** (8x32 architecture). Quantitative comparison of Chamfer Distance (CD) and IoU for shape representation across different architectures and optimizers. CD values are multiplied by $10^4$. Results are mean $\pm$ std over 3 seeds. The best-performing optimizer per row is highlighted in **bold**.

| | Model | CD ($\downarrow$) | | IoU ($\uparrow$) | |
|---|---|---|---|---|---|
| | | Adam | Muon | Adam | Muon |
| Armadillo | ReLU MLP | 0.344 $_{\pm .029}$ | **0.105** $_{\pm .026}$ | 0.937 $_{\pm .003}$ | **0.972** $_{\pm .003}$ |
| | ReLU +PE | 0.121 $_{\pm .041}$ | **0.049** $_{\pm .001}$ | 0.966 $_{\pm .005}$ | **0.979** $_{\pm .001}$ |
| | SIREN | 0.403 $_{\pm .312}$ | **0.039** $_{\pm .001}$ | 0.940 $_{\pm .037}$ | **0.985** $_{\pm .001}$ |
| | FINER | 0.045 $_{\pm .001}$ | **0.039** $_{\pm .001}$ | 0.982 $_{\pm .001}$ | **0.985** $_{\pm .000}$ |
| Thai Statue | ReLU MLP | 0.817 $_{\pm .037}$ | **0.547** $_{\pm .155}$ | 0.785 $_{\pm .025}$ | **0.849** $_{\pm .006}$ |
| | ReLU +PE | 0.451 $_{\pm .028}$ | **0.344** $_{\pm .046}$ | 0.841 $_{\pm .003}$ | **0.880** $_{\pm .004}$ |
| | SIREN | 0.726 $_{\pm .049}$ | **0.086** $_{\pm .012}$ | 0.814 $_{\pm .016}$ | **0.938** $_{\pm .001}$ |
| | FINER | 0.101 $_{\pm .006}$ | **0.068** $_{\pm .005}$ | 0.932 $_{\pm .004}$ | **0.944** $_{\pm .002}$ |

is particularly relevant in light of recent work (Davies et al., 2020; Coiffier & Béthune, 2024), which highlights neural distance fields as a promising alternative to traditional shape representations such as triangle meshes. Following the experimental setups of Davies et al. (2020), we adopt a comparable training protocol and report details of our setup in Appendix B.1. We evaluate our models on two standard benchmarks, the Armadillo and Thai Statue datasets, from the Stanford 3D Scanning Repository (Curless & Levoy, 1996) and report scores in Table 3, with qualitative examples shown in Figure 4.

When training with the orthogonalizing optimizer, we see relevant performance gains in both Chamfer Distance (CD) and Intersection-over-Union (IoU), mirroring the improvements observed for the previous modalities. These gains are notable given the challenging setup of using small, 8×32 architectures. Interestingly, adding positional encodings (PE) yields larger performance gains when training with Adam, suggesting a modeling-capacity gap in these small architectures.

### 4.2. Inverse Problems

#### 4.2.1. CT FROM UNDERSAMPLED MEASUREMENTS

Reconstructing images from highly undersampled measurements, such as in sparse-view computed tomography (CT), constitutes a classical ill-posed inverse problem. Following the setup of Saragadam et al. (2023), we simulate a sparse-view scenario by generating a sinogram with only 100 projections from a $435 \times 326$ ground-truth grayscale thoracic X-ray provided in Clark et al. (2013). An INR is then trained from random initialization to recover an image consistent with these limited measurements. Details of the experimental setup and hyperparameter search configuration are provided in Appendix B.4. Quantitative and qualitative results are presented in Table 4 and Figure 3, respectively. INRs optimized with orthogonalization achieve a higher signal-to-noise ratio (SNR) in regions containing heterogeneous structures, such as the bronchi, as shown in Figure 3 (*red boxes*). Mirroring the results for the signal representa-

tion experiments in Section 4.1, orthogonalized optimization yields the most improvements for ReLU-based INRs. Notably, Muon also generally improves perceptual quality (LPIPS) across all architectures, which is particularly impressive given the relevance and intrinsic ill-posedness of the CT reconstruction problem.

#### 4.2.2. SINGLE IMAGE SUPER-RESOLUTION

We apply our approach to the single-image super-resolution (SISR) task using the DIV2K dataset (Agustsson & Timofte, 2017; Timofte et al., 2017) with an upsampling factor of 4. The reconstruction is performed by representing the high-resolution image implicitly through an INR, enabling the recovery of fine details lost during downsampling. Similar to the CT reconstruction experiment, INRs optimized with Muon demonstrate strong performance on this inverse problem. Notably, as shown in Table 5, orthogonalized optimization substantially improves results for MLPs with ReLU-based activations, making the ReLU FFN particularly effective for this task. Muon does not consistently enhance reconstruction quality for architectures with strong high-frequency inductive biases, such as FINER and SIREN. However, these architectures also no longer outperform the ReLU FFN trained with Muon. We postulate that this is due to the advantageous low-frequency bias of ReLU-based MLPs for this task, which reduces high-frequency artifacts while, through orthogonalized optimization, still retaining the capacity to model fine-scale structures.

#### 4.2.3. NEURAL RADIANCE FIELDS

Finally, we investigate the effect of orthogonal optimization on novel view synthesis (Mildenhall et al., 2021), one of the cornerstone applications of INRs. We closely follow the experimental setup of Liu et al. (2024) and use the synthetic NeRF dataset introduced in the original NeRF paper (Mildenhall et al., 2021). Representative models,

*Table 4.* **Sparse-view CT reconstruction performance.** Quantitative comparison of CT reconstruction quality for different architectures using the Adam optimizer versus the Muon optimizer. We report the mean and standard deviation for PSNR, SSIM, and LPIPS over 5 seeds. The best-performing optimizer for each architecture is highlighted in **bold**.

| Model | PSNR (↑) | | SSIM (↑) | | LPIPS (↓) | |
| --- | --- | --- | --- | --- | --- | --- |
| | Adam | Muon | Adam | Muon | Adam | Muon |
| ReLU MLP | $25.86_{\pm 0.35}$ | $\mathbf{32.93}_{\pm 0.31}$ | $0.640_{\pm 0.014}$ | $\mathbf{0.831}_{\pm 0.004}$ | $0.528_{\pm 0.026}$ | $\mathbf{0.251}_{\pm 0.005}$ |
| ReLU FFN | $28.64_{\pm 0.60}$ | $\mathbf{33.28}_{\pm 0.27}$ | $0.756_{\pm 0.025}$ | $\mathbf{0.839}_{\pm 0.003}$ | $0.347_{\pm 0.058}$ | $\mathbf{0.156}_{\pm 0.010}$ |
| Gauss MLP | $26.01_{\pm 0.50}$ | $\mathbf{27.35}_{\pm 0.43}$ | $0.483_{\pm 0.031}$ | $\mathbf{0.524}_{\pm 0.023}$ | $0.405_{\pm 0.026}$ | $\mathbf{0.340}_{\pm 0.026}$ |
| Gauss FFN | $18.76_{\pm 1.03}$ | $\mathbf{20.81}_{\pm 0.18}$ | $0.177_{\pm 0.035}$ | $\mathbf{0.225}_{\pm 0.006}$ | $0.883_{\pm 0.074}$ | $\mathbf{0.763}_{\pm 0.014}$ |
| SIREN | $30.58_{\pm 0.43}$ | $\mathbf{33.69}_{\pm 0.12}$ | $0.803_{\pm 0.009}$ | $\mathbf{0.850}_{\pm 0.002}$ | $0.316_{\pm 0.011}$ | $\mathbf{0.225}_{\pm 0.003}$ |
| WIRE | $26.26_{\pm 0.36}$ | $\mathbf{29.51}_{\pm 0.72}$ | $0.471_{\pm 0.019}$ | $\mathbf{0.644}_{\pm 0.046}$ | $0.432_{\pm 0.017}$ | $\mathbf{0.251}_{\pm 0.039}$ |
| FINER | $31.93_{\pm 0.50}$ | $\mathbf{32.92}_{\pm 0.21}$ | $0.820_{\pm 0.016}$ | $\mathbf{0.822}_{\pm 0.006}$ | $0.251_{\pm 0.019}$ | $\mathbf{0.215}_{\pm 0.005}$ |

*Table 5.* **Single-image super-resolution performance.** Quantitative comparison of SISR task on DIV2K (`0001.png`) for different architectures using Adam vs. Muon optimizer. We report mean $\pm$ standard deviation over $n = 5$ seeds for PSNR, SSIM, and LPIPS. The best-performing optimizer for each architecture is highlighted in **bold**.

| Model | PSNR (↑) | | SSIM (↑) | | LPIPS (↓) | |
| --- | --- | --- | --- | --- | --- | --- |
| | Adam | Muon | Adam | Muon | Adam | Muon |
| ReLU MLP | $20.71_{\pm 0.16}$ | $\mathbf{23.50}_{\pm 0.05}$ | $0.386_{\pm 0.004}$ | $\mathbf{0.473}_{\pm 0.003}$ | $0.850_{\pm 0.005}$ | $\mathbf{0.743}_{\pm 0.004}$ |
| ReLU FFN | $25.77_{\pm 0.04}$ | $\mathbf{26.82}_{\pm 0.02}$ | $0.618_{\pm 0.004}$ | $\mathbf{0.674}_{\pm 0.002}$ | $0.506_{\pm 0.011}$ | $\mathbf{0.384}_{\pm 0.004}$ |
| Gauss MLP | $23.82_{\pm 0.04}$ | $\mathbf{25.75}_{\pm 0.08}$ | $0.503_{\pm 0.007}$ | $\mathbf{0.592}_{\pm 0.005}$ | $0.545_{\pm 0.004}$ | $\mathbf{0.414}_{\pm 0.009}$ |
| Gauss FFN | $13.86_{\pm 0.18}$ | $\mathbf{14.73}_{\pm 0.00}$ | $0.063_{\pm 0.004}$ | $\mathbf{0.282}_{\pm 0.000}$ | $1.036_{\pm 0.044}$ | $\mathbf{0.894}_{\pm 0.001}$ |
| SIREN | $26.38_{\pm 0.07}$ | $\mathbf{26.41}_{\pm 0.06}$ | $\mathbf{0.650}_{\pm 0.006}$ | $0.638_{\pm 0.004}$ | $0.517_{\pm 0.009}$ | $\mathbf{0.508}_{\pm 0.005}$ |
| WIRE | $25.26_{\pm 0.09}$ | $\mathbf{26.14}_{\pm 0.09}$ | $0.572_{\pm 0.007}$ | $\mathbf{0.627}_{\pm 0.006}$ | $0.547_{\pm 0.006}$ | $\mathbf{0.481}_{\pm 0.008}$ |
| FINER | $\mathbf{26.83}_{\pm 0.05}$ | $26.59_{\pm 0.07}$ | $\mathbf{0.678}_{\pm 0.003}$ | $0.641_{\pm 0.006}$ | $0.476_{\pm 0.006}$ | $\mathbf{0.456}_{\pm 0.008}$ |

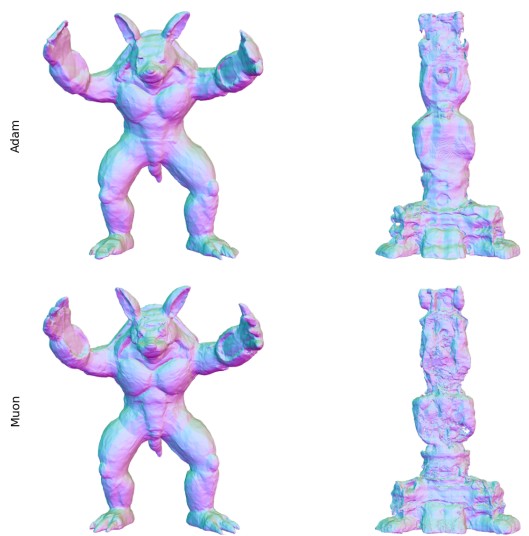

*Figure 4.* Qualitative comparison for Armadillo and Thai Statue from the Stanford 3D Repository (Curless & Levoy, 1996) for ReLU +PE trained with Adam (top) and Muon (bottom).

including the widely adopted Instant-NGP (Müller et al., 2022), are evaluated, with full experimental details provided in Appendix B.6. Once again, we observe consistent and substantial improvements when optimizing neural radiance fields with Muon, consistently improving over the same

architectures. Remarkably, orthogonal optimization reduces reconstruction artifacts (*as visible in Figure 5 for FINER*), resulting in more faithful scene reconstructions, as reflected in both intensity-based and perceptual quality metrics (*see Table 6 and Table 16, and Appendix Table 17 respectively*). This is particularly relevant in novel view synthesis, where mitigating visual artifacts arising from incorrect density estimates remains an active area of study.

## 5. Limitations

While our results identify stable rank as a novel, unifying framework spanning input-, activation-, and optimization-level interventions, we acknowledge several limitations.

First, our stable rank framework provides a useful diagnostic, but it does not predict an optimal target rank. As shown, maximal stable rank may not be desirable in all settings. For example, for ill-posed inverse problems, a low induced stable rank may act as an implicit regularizer, particularly for expressive INR architectures. A full characterization of how the different interventions interact across signal classes, architectures, and tasks, therefore, remains an important direction for future work. While optimization-level interventions consistently improve every architecture we study, they narrow but do not fully close the gap to more expressive architectures, e.g., a ReLU MLP trained with Muon still remains less expressive than SIREN or FINER. Our analysis

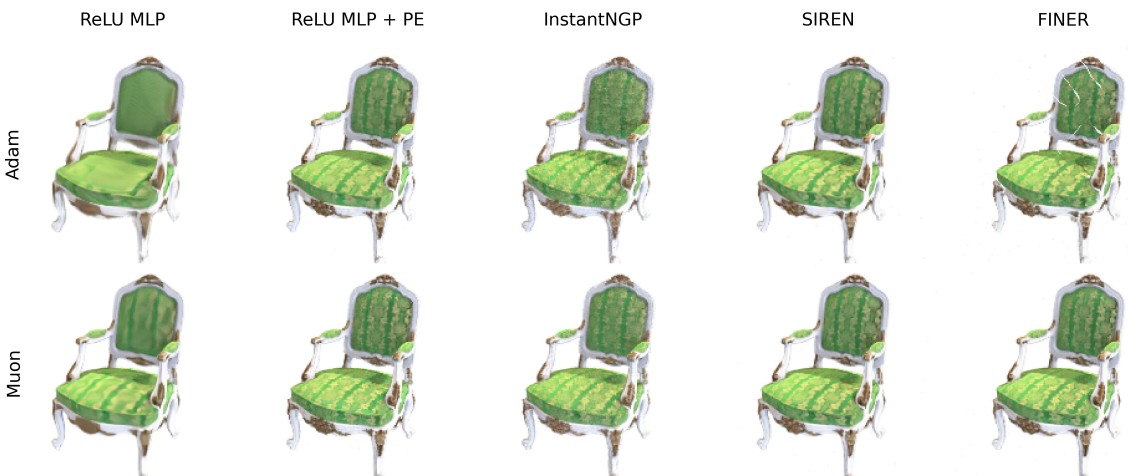

*Figure 5.* Reconstructing a view from a neural radiance field depicting a chair, optimized with Adam (top row) and Muon (bottom row). Rendered videos of a 360-degree view are provided on the project website.

*Table 6.* **NeRF performance (PSNR)**. Quantitative comparison of NeRF reconstruction quality across different architectures using the Adam and Muon optimizers. We report the mean $\pm$ standard deviation over 3 seeds for PSNR. The best-performing optimizer for each architecture is highlighted in **bold**. In a few cases, for scenes marked with $^\dagger$, learning rates from hyperparameter sweeps did not transfer effectively, resulting in non-convergent training, where we resorted to the next stable learning rate in the sweep.

| | PSNR ($\uparrow$) | | | | | | | |
| Methods | Chair | | Drums | | Ficus | | Hotdog | |
| | Adam | Muon | Adam | Muon | Adam | Muon | Adam | Muon |
| ReLU MLP | $29.36_{\pm0.48}$ | $\mathbf{31.73}_{\pm\mathbf{0.16}}$ | $18.79_{\pm1.00}$ | $\mathbf{21.84}_{\pm\mathbf{0.06}}$ | $22.67^{\dagger}_{\pm0.53}$ | $\mathbf{24.91}_{\pm\mathbf{0.14}}$ | $28.69_{\pm1.27}$ | $\mathbf{33.17}_{\pm\mathbf{0.30}}$ |
| ReLU MLP+PE | $33.04_{\pm0.05}$ | $\mathbf{33.39}_{\pm\mathbf{0.16}}$ | $22.84_{\pm0.07}$ | $\mathbf{24.29}_{\pm\mathbf{0.05}}$ | $26.74_{\pm0.05}$ | $\mathbf{26.97}_{\pm\mathbf{0.16}}$ | $32.70_{\pm0.12}$ | $\mathbf{33.27}_{\pm\mathbf{0.02}}$ |
| InstantNGP | $29.81_{\pm0.08}$ | $\mathbf{31.39}_{\pm\mathbf{0.09}}$ | $23.63_{\pm0.18}$ | $\mathbf{24.15}_{\pm\mathbf{0.01}}$ | $25.47_{\pm0.04}$ | $\mathbf{25.97}_{\pm\mathbf{0.09}}$ | $27.99^{\dagger}_{\pm0.52}$ | $\mathbf{30.91}_{\pm\mathbf{0.05}}$ |
| SIREN | $33.22_{\pm0.03}$ | $\mathbf{34.17}_{\pm\mathbf{0.04}}$ | $24.65_{\pm0.04}$ | $\mathbf{24.96}_{\pm\mathbf{0.07}}$ | $27.80_{\pm0.05}$ | $\mathbf{28.27}_{\pm\mathbf{0.06}}$ | $33.31_{\pm0.33}$ | $\mathbf{33.86}_{\pm\mathbf{0.11}}$ |
| FINER | $32.40_{\pm0.89}$ | $\mathbf{34.22}_{\pm\mathbf{0.17}}$ | $24.60_{\pm0.12}$ | $\mathbf{24.87}_{\pm\mathbf{0.11}}$ | $\mathbf{28.15}_{\pm\mathbf{0.13}}$ | $28.14_{\pm0.30}$ | $32.36_{\pm1.60}$ | $\mathbf{33.88}_{\pm\mathbf{0.09}}$ |

also bounds the stable rank of each weight update through the span of its input activations rather than fully characterizing the optimization trajectory; a deeper theoretical account of how orthogonalized updates shape this trajectory is left to future work.

Second, we focus on Muon as a representative orthogonalizing optimizer. Although the Newton-Schulz approximation efficiently approximates rank-preserving updates, we do not study the implications of this approximation in detail, and believe Muon should be viewed as one point in a broader design space rather than a final, optimal solution. Softer or adaptive forms of orthogonalization, or combinations with curvature-aware preconditioning may yield better trade-offs between expressivity, smoothness, and efficiency. Recent progress on matrix sign and polar-factor approximations (Amsel et al., 2025) may further improve the scalability of such methods, and the design of novel rank-preserving regularizers offer a complementary alternative.

Finally, Muon introduces additional tuning complexity through its learning-rate schedules, which we note as a particularly practical relevant limitation worth exploring.

## 6. Discussion and Future Work

In this work, we shift focus from architectural design to optimization and rank preservation, highlighting that the choice of optimizer remains a severely underexplored aspect when learning implicit neural representations, regardless of the underlying architecture. Our findings demonstrate that optimization alone can significantly impact model fidelity, challenging the prevailing perception that architectural changes are the primary driver of INR performance. This is particularly striking for vanilla ReLU MLPs, which have long been believed incapable of representing signals with high fidelity. More broadly, our results establish optimization as a third axis of INR design, complementary to input- and activation-level interventions. Viewed through the lens of stable rank, these axes act additively, and combining them yields the most faithful reconstructions.

Looking forward, we believe INRs offer a clean testbed for studying how architecture and optimization jointly determine spectral bias, and we hope this perspective inspires new directions within and beyond INRs.

# Acknowledgments

This work is funded by the Munich Center for Machine Learning. Julian McGinnis and Mark Mühlau are supported by Bavarian State Ministry for Science and Art (Collaborative Bilateral Research Program Bavaria – Quebec: AI in medicine, grant F.4-V0134.K5.1/86/34). Suprosanna Shit is supported by the UZH Postdoc Grant (K-74851-03-01). Mark Mühlau is supported by the German Research Foundation (DFG) under the SPP Radiomics project number 428223038. Suprosanna Shit and Bjoern Menze acknowledge support by the Helmut Horten Foundation.

# Impact Statement

This paper presents work whose goal is to advance the field of Machine Learning. There are many potential societal consequences of our work, none which we feel must be specifically highlighted here.

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

# A. Optimizer Setup

**General Setup:**  We compare Muon's high-rank, near-orthogonal with conventional Adam-based optimization for INRs across several established tasks. For Adam, we follow standard practice and use a single learning rate for all parameter groups. For Muon, we adopt the hybrid optimization strategy described by Jordan et al. (2024): hidden-layer weight matrices are optimized with Muon, while the input/output weight matrices and all bias parameters are optimized with Adam. This results in two parameter groups with (potentially) different learning rates. Unless stated otherwise, we refer to the learning rate used by standalone Adam as *Adam LR*. For Muon-based training, we use *Muon LR* for the Muon-optimized parameters and *Aux. LR* for the Adam-optimized parameter group. We set $\beta = (0.9, 0.999)$ for both optimizers, following the default PyTorch settings.

To isolate the effect of orthogonal optimization from additional forms of regularization, we use Muon in its vanilla configuration, i.e., without the weight decay term proposed by Liu et al. (2025), since weight decay is typically not applied when training INRs with Adam. We employ learning rate scheduling that follows domain-specific standards: we use cosine annealing for audio, image, and CT reconstruction, and exponential decay for NeRF and 3D shape reconstruction. For fairness, both Adam and Muon employ the same scheduling strategy within each experimental setting.

**Learning Rate Sweep:**  Unless otherwise noted, we sweep learning rates over $AdamLR = \{1 \cdot 10^{-5}, 3 \cdot 10^{-5}, 1 \cdot 10^{-4}, 3 \cdot 10^{-4}, \dots, 1 \cdot 10^{-1}\}$. For Muon-INRs, we similarly perform a grid search over both Muon LR and the auxiliary Adam LR to ensure optimizer fairness: $MuonLR = \{1 \cdot 10^{-5}, 5 \cdot 10^{-5}, 1 \cdot 10^{-4}, 5 \cdot 10^{-4}, \dots, 1 \cdot 10^{-1}\}$ and Adam learning rates $Aux.LR = \{1 \cdot 10^{-5}, 5 \cdot 10^{-5}, 1 \cdot 10^{-4}, \dots, 1 \cdot 10^{-2}\; 1 \cdot 10^{-1}\}$, resulting in 81 configurations per architecture ($9 \cdot 9 = 81$).

# B. Experimental Details

## B.1. Image Overfitting

**Experimental Setup:**  We closely follow the image overfitting setup of Saragadam et al. (2023). All experiments employ a 5-layer MLP (comprising three hidden layers between the input and output) with 300 hidden dimensions, except for INRs, which use an additional encoding, where a 4-layer MLP is employed instead. For the complex-valued WIRE, we adopt complex-valued layers, resulting in an effective width of $300/\sqrt{2} \approx 212$. For the real-valued WIRE, we retain a hidden dimension of 300. Each model is trained for 5000 epochs, and test metrics are logged every 500 epochs. Cosine annealing is used for learning rate scheduling following (Ma et al., 2024), with a minimum learning rate of $10^{-6}$ across all optimizers. Following Saragadam et al. (2023), architecture-specific hyperparameters are set as in prior work: $\sigma = 10$ for FFN (Tancik et al., 2020); $\omega_0 = 40$ and $\omega_{\text{hidden}} = 40$ for SIREN and FINER (Sitzmann et al., 2020b; Liu et al., 2024); scale $= 30$ for Gaussian activations (Ramasinghe & Lucey, 2022); and $\sigma = 10, \omega = 20$ for WIRE (Saragadam et al., 2023). Following Liu et al. (2024), we initialize the bias vector uniformly in the range $[-b, b]$, where $b = 1/\sqrt{2}$. We adopt the Gaussian activation from Ramasinghe & Lucey (2022), defined as

$$\sigma(x) = e^{-\frac{0.5x^2}{a^2}},$$

which is equivalent to the WIRE activation $\sigma(x) = e^{-(s_0 x)^2}$ when $\frac{0.5}{a^2} = s_0^2$, yielding

$$a = \frac{1}{\sqrt{2}s_0}.$$

Following the setup of (Saragadam et al., 2023) using $s_0 = 30$, this gives $a = \frac{\sqrt{0.5}}{30} \approx 0.0236$, which we use subsequently set in our image experiments.

**Hyperparameter Search:**  We conduct the learning rate grid search on the first image of the Kodak dataset. All models, both during hyperparameter sweeps and final evaluations, are trained for the full 5000 epochs. We select the best configuration based on LPIPS (Zhang et al., 2018), as it correlates best with perceptual image quality, which motivates us to use it as a selection criterion throughout all image-based experiments. Subsequently, we use the best-performing learning rate configuration across the remaining dataset (23 images) and report the metrics for the full dataset. We report the employed learning rates and attained scores in Table 7.

## B.2. Audio Overfitting

**Experimental Setup:**  Following (Sitzmann et al., 2020b; Essakine et al., 2025), we normalize the temporal coordinates to the range $[-100, 100]$ for sinusoidal-based INRs (**SIREN** and **FINER**), which may also be seen as a frequency scaling factor for the first $\omega$. Similar to (Essakine et al., 2025), we retain the standard temporal range $[-1, 1]$ for all other architectures. All models are trained for 5000 epochs.

**Hyperparameter Search:**  We perform grid searches for both Adam and Muon independently for each audio file and select the configuration that yields the best results for each model. We report the employed learning rates and attained scores in Table 7.

We report the reconstruction results for Bach in the main manuscript and include additional results for *counting.wav* (Sitzmann et al., 2020b) in section C.3.

*Table 7.* Overview of the best learning rate configurations for *kodim01* selected by the lowest attained LPIPS (↓) score.

| Model | Adam | | | | Muon | | | | |
|---|---|---|---|---|---|---|---|---|---|
| | Adam LR | PSNR (↑) | SSIM (↑) | LPIPS (↓) | Aux. LR | Muon LR | PSNR (↑) | SSIM (↑) | LPIPS (↓) |
| ReLU MLP | 0.0030 | 20.17 | 0.329 | 0.7893 | 0.0030 | 0.1000 | 26.30 | 0.726 | 0.337 |
| ReLU FFN | 0.0030 | 28.68 | 0.840 | 0.1385 | 0.0100 | 0.1000 | 36.94 | 0.965 | 0.018 |
| Gaussian MLP | 0.0030 | 33.77 | 0.943 | 0.0334 | 0.0030 | 0.0100 | 38.75 | 0.984 | 0.013 |
| Gaussian FFN | 0.0100 | 26.51 | 0.820 | 0.1381 | 0.0010 | 0.0100 | 30.90 | 0.906 | 0.058 |
| SIREN | 0.0010 | 36.96 | 0.970 | 0.0155 | 0.0010 | 0.0030 | 39.92 | 0.984 | 0.005 |
| WIRE (real-valued) | 0.0100 | 38.88 | 0.985 | 0.0132 | 0.0003 | 0.0100 | 40.23 | 0.987 | 0.008 |
| WIRE (complex-valued) | 0.0030 | 33.78 | 0.947 | 0.0437 | 0.0030 | 0.0100 | 27.96 | 0.862 | 0.106 |
| FINER | 0.0003 | 37.88 | 0.974 | 0.0089 | 0.0030 | 0.0010 | 40.74 | 0.986 | 0.004 |

*Table 8.* Comparison of audio results for reconstructing *bach.wav* for each architecture, optimized with Adam vs. Muon. The best runs are selected based on the best attained SNR (↑).

| Model | Adam | | | Muon | | | |
|---|---|---|---|---|---|---|---|
| | Adam LR | Full-SNR (↑) | SI-SNR (↑) | Aux. LR | Muon LR | Full-SNR (↑) | SI-SNR (↑) |
| ReLU MLP | 0.000 300 | 0.02 | −23.52 | 0.000 100 | 0.010 000 | 0.14 | −14.88 |
| ReLU FFN | 0.001 000 | 8.88 | 8.31 | 0.003 000 | 0.100 000 | 21.43 | 21.40 |
| Gaussian MLP | 0.001 000 | 8.76 | 8.16 | 0.001 000 | 0.003 000 | 12.85 | 12.64 |
| Gaussian FFN | 0.001 000 | 37.80 | 37.80 | 0.001 000 | 0.003 000 | 46.80 | 46.80 |
| SIREN | 0.000 300 | 37.92 | 37.92 | 0.000 300 | 0.003 000 | 47.46 | 47.46 |
| WIRE (real-valued) | 0.010 000 | 3.65 | 1.23 | 0.003 000 | 0.030 000 | 15.52 | 15.62 |
| FINER | 0.000 100 | 27.22 | 27.21 | 0.000 100 | 0.001 000 | 36.48 | 36.48 |

*Table 9.* Comparison of audio results for reconstructing *counting.wav* for each architecture, optimized with Adam vs. Muon. The best runs are selected based on the best attained SNR (↑).

| Model | Adam | | | Muon | | | |
|---|---|---|---|---|---|---|---|
| | Adam LR | Full-SNR (↑) | SI-SNR (↑) | Aux. LR | Muon LR | Full-SNR (↑) | SI-SNR (↑) |
| ReLU MLP | 0.000 300 | 0.00 | −45.75 | 0.000 030 | 0.010 000 | 0.00 | −44.53 |
| ReLU FFN | 0.003 000 | 1.37 | −4.32 | 0.000 300 | 0.030 000 | 9.66 | 9.17 |
| Gaussian MLP | 0.001 000 | 1.64 | −3.23 | 0.001 000 | 0.003 000 | 4.22 | 2.29 |
| Gaussian FFN | 0.000 300 | 18.30 | 18.23 | 0.000 300 | 0.001 000 | 18.31 | 18.25 |
| SIREN | 0.001 000 | 17.59 | 17.52 | 0.000 300 | 0.003 000 | 25.38 | 25.38 |
| WIRE (real-valued) | 0.010 000 | 0.16 | −14.15 | 0.010 000 | 0.030 000 | 1.56 | −3.32 |
| FINER | 0.000 100 | 22.09 | 22.07 | 0.000 100 | 0.001 000 | 28.19 | 28.19 |

### B.3. Signed Distance Field Overfitting

**Experimental Setup:** For shape reconstruction tasks, we employ a model with eight hidden layers and a hidden dimension of 32, identical to the (blue) default setup of (Davies et al., 2020). We employ a sampling strategy based on the zero level set, where Laplacian noise is introduced to the sampled locations (Davies et al., 2020), resulting in an exponential decay in sample density away from the surface. Following Lindell et al. (2022), we adopt a dual-scale approach with fine samples ($\sigma_L^2 = 2 \times 10^{-6}$) and coarse samples ($\sigma_L^2 = 2 \times 10^{-2}$). We use a weighted loss function to balance the contributions from fine and coarse samples:

$$\mathcal{L}_{\text{SDF}} = \lambda_{\text{SDF}} \|\mathbf{y}^c - \mathbf{y}_{\text{GT}}^c\|_2^2 + \|\mathbf{y}^f - \mathbf{y}_{\text{GT}}^f\|_2^2, \quad (4)$$

where $\mathbf{y}^f$ and $\mathbf{y}^c$ denote the network outputs for fine and coarse samples, respectively, $\mathbf{y}_{\text{GT}}$ are the ground truth SDF values, and $\lambda_{\text{SDF}} = 0.01$.

**Hyperparameter Sweep:** Given the comparatively long training times for shape models, we conduct a learning rate hyperparameter search using 20,000 iterations on the Armadillo shape and then proceed to train the final models with the lowest attained total loss for a total of 200,000 iterations (from scratch). We report the employed learning rates in Table 10.

### B.4. CT Reconstruction

**Experimental Setup:** Following (Saragadam et al., 2023; Essakine et al., 2025), we use the identical image from The Cancer Imaging Archive (TCIA) (Clark et al., 2013) and adopt the model architectures specified in (Tancik et al., 2020; Saragadam et al., 2023) for medical images. The baseline architecture for all INRs is a 5-layer MLP with a hidden dimension of 256. The exceptions are ReLU FFN and Gaussian FFN, which use 4 layers due to their 256-dimensional input embedding. We use the following specific hyperparameters for the different INR models: $\sigma = 4.0$ for Fourier Features (sampled from a Gaussian distribution per (Tancik et al., 2020)); $w_0 = 30$ and $w_{\text{hidden}} = 50$ for SIREN and FINER; $scale = 0.05$ for Gaussian activations; and $\omega = 10.0$ with $\sigma = 10.0$ for WIRE. Following (Saragadam et al., 2023), all models are trained for 5000 epochs.

*Table 10.* Best learning-rate configurations (selected by lowest 20k-step total loss) for each architecture optimizing a neural representation of the *Armadillo*, under Adam versus Muon. For Muon, **Aux. LR** is the auxiliary Adam learning rate (first/last layers, biases) and **Muon LR** is the learning rate for the orthogonalized hidden-weight updates.

| Model | Adam | | | | Muon | | | | |
|---|---|---|---|---|---|---|---|---|---|
| | Adam LR | Total Loss ($\downarrow$) | Fine Loss ($\downarrow$) | Coarse Loss ($\downarrow$) | Aux. LR | Muon LR | Total Loss ($\downarrow$) | Fine Loss ($\downarrow$) | Coarse Loss ($\downarrow$) |
| ReLU MLP | 0.0030 | 0.009 60 | 0.007 56 | 0.002 04 | 0.0030 | 0.0100 | 0.006 00 | 0.004 89 | 0.001 12 |
| ReLU MLP + PE | 0.0030 | 0.007 80 | 0.006 07 | 0.001 73 | 0.0100 | 0.0100 | 0.005 64 | 0.004 27 | 0.001 37 |
| SIREN | 0.0010 | 0.005 14 | 0.003 82 | 0.001 32 | 0.0003 | 0.0010 | 0.003 12 | 0.002 07 | 0.001 04 |
| FINER | 0.0003 | 0.006 45 | 0.004 93 | 0.001 52 | 0.0010 | 0.0003 | 0.003 26 | 0.002 35 | 0.000 91 |

Regarding the loss, we minimize the L2-norm data fidelity loss in the measurement space:

$$\mathcal{L}(\theta) = \|\mathcal{A}(f_\theta) - y\|_2^2$$

where $\mathcal{A}$ is the forward Radon transform operator. By holding all architecture-specific hyperparameters constant, this setup allows us to attribute differences in final reconstruction quality directly to the convergence properties and implicit regularization of the optimizers under investigation.

**Hyperparameter Sweeps:** We first sweep Adam learning rates over $LR_{\text{Adam}} = \{1 \times 10^{-5}, 5 \times 10^{-5}, 1 \times 10^{-4}, 5 \times 10^{-4}, \dots, 1 \times 10^{-2}\}$ to establish a baseline for each architecture. For Muon-INRs, we perform a fully exhaustive grid search to ensure optimizer fairness, using Muon learning rates $LR_{\text{Muon}} = \{1 \times 10^{-5}, 5 \times 10^{-5}, 1 \times 10^{-4}, 5 \times 10^{-4}, \dots, 1 \times 10^{-1}\}$ and auxiliary Adam learning rates $LR_{\text{Adam, Muon}} = \{1 \times 10^{-5}, 5 \times 10^{-5}, 1 \times 10^{-4}, 5 \times 10^{-4}, \dots, 1 \times 10^{-2}\}$, yielding 63 configurations per architecture ($9 \times 7 = 63$).

### B.5. Single Image Super Resolution

**Experimental Setup:** We perform single-image super-resolution (SISR) experiments using the first image (0001.png) from the DIV2K dataset. Consistent with (Essakine et al., 2025), we formalize the task in the following way:

Let $x_{\text{HR}} \in \mathbb{R}^{H \times W}$ denote the underlying high-resolution image. We generate the corresponding low-resolution observation $x_{\text{LR}}$ by applying a downsampling operator $D$, such that $x_{\text{LR}} = D(x_{\text{HR}})$, where $D$ reduces the spatial resolution by a factor of four. For reconstruction, we parameterize the estimated high-resolution signal using an INR $f_\theta : \mathbb{R}^2 \to \mathbb{R}$. The model *is trained by enforcing consistency between the downsampled INR output and the observed low-resolution image, i.e., $\theta^\star = \arg\min_\theta \mathcal{L}(D(f_\theta), x_{\text{LR}})$, where $\mathcal{L}$ denotes the pixel-wise reconstruction loss. After optimization, the super-resolved image is obtained by evaluating $f_{\theta^\star}$ on the high-resolution spatial grid.

We closely follow the SISR setup of Saragadam et al. (2023); Essakine et al. (2025). All experiments employ a 5-layer MLP (comprising three hidden layers between the input

and output) with 256 hidden dimensions, except for INRs, which use an additional encoding, where a 4-layer MLP is employed instead. Following Saragadam et al. (2023), architecture-specific hyperparameters are set as : $\sigma = 10$ for FFN (Tancik et al., 2020); $\omega_0 = 30$, $\omega_{\text{hidden}} = 50$, and hidden layer $\omega = 50$ for SIREN and FINER (Sitzmann et al., 2020b; Liu et al., 2024); scale $= 0.05$ for Gaussian activations (Ramasinghe & Lucey, 2022); and $\sigma = 6.0$, $\omega = 8.0$ for WIRE (Saragadam et al., 2023).

**Hyperparameter Search:** We perform a grid search using the full epoch duration of 5000 epochs. We report the best reconstruction results in the main script, and complement this table with the utilized learning rates in Table 12.

### B.6. Neural Radiance Fields

**Experimental Setup:** We base our experiments on the PyTorch implementations (Quei-An, 2020; Tang, 2022; Tang et al., 2022) of Instant-NGP (Müller et al., 2022), which allows us to seamlessly integrate Muon into an existing NeRF training pipeline. NeRFs (Mildenhall et al., 2021) model a 3D scene as a continuous volumetric function parameterized by an MLP, mapping a spatial coordinate $(x, y, z)$ and viewing direction $(\theta, \phi)$ to volume density and emitted radiance. Novel views are rendered by sampling points along camera rays, querying the MLP at each location, and applying differentiable volumetric integration. Following Saragadam et al. (2023); Liu et al. (2024), we train NeRFs on a challenging, sparse-view setting using a subset of 25 images (from the original 100). Following (Liu et al., 2024), we use 6-layer MLPs with a hidden dimension of 128. Moreover, following (Tang, 2022), we use float-16bit training for both optimizers.

**Hyperparameter Search:** We first perform a hyperparameter sweep on the `ship` dataset, and subsequently train final models for 37,500 iterations across all datasets using the best lr config from ship.

*Table 11.* Comparison of best runs for each architecture in the CT reconstruction experiment, optimized with Adam vs. Muon and selected by lowest LPIPS score ($\downarrow$).

| Model | Adam | | | | Muon | | | | |
|---|---|---|---|---|---|---|---|---|---|
| | Adam LR | PSNR ($\uparrow$) | SSIM ($\uparrow$) | LPIPS ($\downarrow$) | Aux. LR | Muon LR | PSNR ($\uparrow$) | SSIM ($\uparrow$) | LPIPS ($\downarrow$) |
| ReLU MLP | 0.0050 | 26.31 | 0.655 | 0.4930 | 0.0050 | 0.0500 | 32.94 | 0.831 | 0.249 |
| ReLU FFN | 0.0050 | 29.36 | 0.785 | 0.2820 | 0.0100 | 0.0500 | 33.24 | 0.812 | 0.117 |
| Gaussian MLP | 0.0100 | 26.63 | 0.518 | 0.3750 | 0.0100 | 0.0100 | 28.10 | 0.566 | 0.290 |
| Gaussian FFN | 0.0050 | 20.08 | 0.213 | 0.7870 | 0.0010 | 0.0100 | 20.58 | 0.218 | 0.775 |
| SIREN | 0.0005 | 29.17 | 0.644 | 0.2960 | 0.0010 | 0.0005 | 33.09 | 0.803 | 0.145 |
| WIRE (real-valued) | 0.0010 | 26.30 | 0.475 | 0.4290 | 0.0100 | 0.0100 | 28.90 | 0.609 | 0.285 |
| WIRE (complex-valued) | 0.0050 | 26.25 | 0.494 | 0.3900 | 0.0050 | 0.0100 | 27.84 | 0.581 | 0.275 |
| FINER | 0.0001 | 32.13 | 0.830 | 0.2440 | 0.0010 | 0.0005 | 31.65 | 0.756 | 0.164 |

*Table 12.* Comparison of best runs for each architecture in the SISR experiment, optimized with Adam vs. Muon and selected by lowest LPIPS score ($\downarrow$). For two configurations (Adam ReLU FFN, Muon ReLU MLP) the chosen LR is the stability-aware 2nd-best LR by LPIPS, as the LPIPS-best LR diverged under multiple seeds.

| Model | Adam | | | | Muon | | | | |
|---|---|---|---|---|---|---|---|---|---|
| | Adam LR | PSNR ($\uparrow$) | SSIM ($\uparrow$) | LPIPS ($\downarrow$) | Aux. LR | Muon LR | PSNR ($\uparrow$) | SSIM ($\uparrow$) | LPIPS ($\downarrow$) |
| ReLU MLP | 0.003 00 | 20.85 | 0.389 | 0.8515 | 0.003 00 | 0.100 00 | 23.49 | 0.474 | 0.746 |
| ReLU FFN | 0.003 00 | 25.82 | 0.623 | 0.4873 | 0.010 00 | 0.100 00 | 26.94 | 0.684 | 0.348 |
| Gaussian MLP | 0.030 00 | 23.80 | 0.505 | 0.5274 | 0.001 00 | 0.030 00 | 25.85 | 0.599 | 0.408 |
| Gaussian FFN | 0.000 01 | 14.08 | 0.066 | 1.0639 | 0.100 00 | 0.000 30 | 14.73 | 0.282 | 0.892 |
| SIREN | 0.001 00 | 26.50 | 0.658 | 0.5060 | 0.010 00 | 0.003 00 | 26.50 | 0.647 | 0.504 |
| WIRE (real-valued) | 0.030 00 | 25.38 | 0.579 | 0.5386 | 0.003 00 | 0.030 00 | 26.22 | 0.634 | 0.476 |
| WIRE (complex-valued) | 0.010 00 | 25.34 | 0.590 | 0.5959 | 0.010 00 | 0.030 00 | 25.64 | 0.612 | 0.506 |
| FINER | 0.000 30 | 26.90 | 0.684 | 0.4644 | 0.000 30 | 0.003 00 | 26.60 | 0.641 | 0.441 |

*Table 13.* Performance comparison of models optimized with Adam versus Muon in the *NeRF Ship* reconstruction experiment. For each optimizer, the best run is selected based on the lowest LPIPS score ($\downarrow$).

| Model | Adam | | | | Muon | | | | |
|---|---|---|---|---|---|---|---|---|---|
| | Adam LR | PSNR($\uparrow$) | SSIM($\uparrow$) | LPIPS ($\downarrow$) | Aux. LR | Muon LR | PSNR($\uparrow$) | SSIM($\uparrow$) | LPIPS ($\downarrow$) |
| ReLU MLP | 0.0100 | 19.7476 | 0.7077 | 0.2693 | 0.0100 | 0.0300 | 21.5647 | 0.7618 | 0.2415 |
| ReLU MLP + PE | 0.0030 | 22.3715 | 0.7806 | 0.1169 | 0.0100 | 0.0300 | 23.5275 | 0.8219 | 0.0880 |
| Instant-NGP | 0.0100 | 19.8986 | 0.6940 | 0.2377 | 0.0001 | 0.0300 | 20.1980 | 0.7385 | 0.1674 |
| SIREN | 0.0003 | 22.4775 | 0.7873 | 0.1237 | 0.0003 | 0.0010 | 22.7716 | 0.8043 | 0.1117 |
| FINER | 0.0003 | 22.3464 | 0.7786 | 0.1243 | 0.0003 | 0.0010 | 22.7079 | 0.7985 | 0.1113 |

*Table 14.* **Image overfitting: interventions vs. the optimizer.** Reconstruction quality and stable rank on a selected animal AFHQ tiger image. We compare a vanilla ReLU MLP trained with Adam, five interventions applied on top of it (batch normalization, rank-inducing pre-training; all with Adam), and the same vanilla ReLU MLP trained with Muon. We report mean and standard deviation for PSNR, SSIM, LPIPS, and the stable rank of the input layer ($SR_{in}$), each hidden layer ($SR_1$, $SR_2$, $SR_3$), and the output layer ($SR_{out}$) over three random seeds.

| Model | PSNR ($\uparrow$) | SSIM ($\uparrow$) | LPIPS ($\downarrow$) | $SR_{in}$ | $SR_1$ | $SR_2$ | $SR_3$ | $SR_{out}$ |
|---|---|---|---|---|---|---|---|---|
| Vanilla ReLU MLP (Adam) | $19.62_{\pm 0.21}$ | $0.561_{\pm 0.013}$ | $0.491_{\pm 0.019}$ | $1.82_{\pm 0.07}$ | $10.52_{\pm 1.03}$ | $8.37_{\pm 1.69}$ | $10.27_{\pm 1.83}$ | $1.24_{\pm 0.04}$ |
| + BatchNorm (pre-act) | $19.71_{\pm 0.32}$ | $0.591_{\pm 0.015}$ | $0.513_{\pm 0.005}$ | $1.83_{\pm 0.07}$ | $66.23_{\pm 1.32}$ | $65.35_{\pm 0.86}$ | $66.07_{\pm 0.66}$ | $2.32_{\pm 0.08}$ |
| + BatchNorm (post-act) | $19.97_{\pm 0.18}$ | $0.589_{\pm 0.009}$ | $0.513_{\pm 0.012}$ | $1.82_{\pm 0.07}$ | $66.34_{\pm 0.59}$ | $65.31_{\pm 0.93}$ | $66.06_{\pm 1.17}$ | $2.38_{\pm 0.13}$ |
| + Pretrain (last) | $20.81_{\pm 0.10}$ | $0.640_{\pm 0.005}$ | $0.419_{\pm 0.008}$ | $1.83_{\pm 0.08}$ | $13.81_{\pm 1.53}$ | $15.56_{\pm 2.01}$ | $13.67_{\pm 2.48}$ | $1.15_{\pm 0.03}$ |
| + Pretrain (all) | $21.58_{\pm 0.24}$ | $0.694_{\pm 0.011}$ | $0.347_{\pm 0.006}$ | $1.83_{\pm 0.06}$ | $15.79_{\pm 0.41}$ | $16.96_{\pm 1.16}$ | $14.95_{\pm 0.77}$ | $1.10_{\pm 0.01}$ |
| + Pretrain (first) | $21.26_{\pm 0.60}$ | $0.666_{\pm 0.043}$ | $0.394_{\pm 0.050}$ | $1.82_{\pm 0.08}$ | $12.12_{\pm 0.73}$ | $16.07_{\pm 0.59}$ | $12.44_{\pm 0.89}$ | $1.09_{\pm 0.03}$ |
| Vanilla ReLU MLP (Muon) | $24.57_{\pm 0.04}$ | $0.852_{\pm 0.002}$ | $0.147_{\pm 0.004}$ | $1.81_{\pm 0.09}$ | $14.53_{\pm 4.27}$ | $44.65_{\pm 2.12}$ | $30.35_{\pm 7.28}$ | $1.19_{\pm 0.06}$ |

## C. Additional Results

### C.1. Rank-Preserving Interventions

In this section, we complement the qualitative results displayed in Figure 2 with quantitative results in Table 14, indicating robustness of the results across multiple runs with different seeds.

Two observations are worth noting. First, Muon notably increases the stable rank in hidden layers to high values, achieving the largest reconstruction improvement (+5dB PSNR).

Also, BatchNorm raises stable rank to near the layer-width ceiling (SR $\approx$ 66 for width-256 hidden layers) but yields only marginal PSNR gains over the vanilla baseline. This

*Table 15.* **Audio overfitting performance (counting.wav).** Mean ± std over 5 seeds. Best-performing optimizer per row in **bold**.

| Model | SNR ($\uparrow$) | | SI-SNR ($\uparrow$) | |
|---|---|---|---|---|
| | Adam | Muon | Adam | Muon |
| ReLU MLP | $0.00_{\pm 0.00}$ | $\mathbf{0.00}_{\pm 0.00}$ | $-45.75_{\pm 0.29}$ | $-\mathbf{44.53}_{\pm 2.02}$ |
| ReLU FFN | $1.37_{\pm 0.24}$ | $\mathbf{9.66}_{\pm 0.17}$ | $-4.32_{\pm 0.90}$ | $\mathbf{9.17}_{\pm 0.19}$ |
| Gauss MLP | $1.64_{\pm 0.15}$ | $\mathbf{4.22}_{\pm 0.26}$ | $-3.23_{\pm 0.51}$ | $\mathbf{2.29}_{\pm 0.40}$ |
| Gauss FFN | $18.30_{\pm 0.17}$ | $\mathbf{18.31}_{\pm 0.35}$ | $18.23_{\pm 0.17}$ | $\mathbf{18.25}_{\pm 0.35}$ |
| SIREN | $17.59_{\pm 1.55}$ | $\mathbf{25.38}_{\pm 0.20}$ | $17.52_{\pm 1.57}$ | $\mathbf{25.38}_{\pm 0.20}$ |
| WIRE | $0.16_{\pm 0.04}$ | $\mathbf{1.56}_{\pm 0.27}$ | $-14.15_{\pm 1.33}$ | $-\mathbf{3.32}_{\pm 0.93}$ |
| FINER | $22.09_{\pm 0.49}$ | $\mathbf{28.19}_{\pm 2.95}$ | $22.07_{\pm 0.49}$ | $\mathbf{28.19}_{\pm 2.96}$ |

indicates that high activation rank is necessary but not sufficient for high-fidelity reconstruction: the geometry of the weight *updates*, not only the activations, determines whether the network can fit high-frequency content. This is consistent with the formulation in Section 3.3, where we argue that optimization-level interventions act on a distinct axis from activation-level ones.

## C.2. Image Overfitting

Due to space constraints, we present representative qualitative results for all model architectures in Figure 7, Figure 8, and Figure 9. Consistent with the quantitative findings, models optimized with Muon show greater relative improvements, particularly for architectures that struggle to capture fine details when trained with Adam, such as ReLU MLP and ReLU FFN. While Muon yields consistent improvements across all models, these gains are less pronounced for architectures like SIREN and FINER, which already achieve high signal fidelity with Adam optimization.

## C.3. Audio Overfitting

We present quantitative results for reconstructing **counting.wav** across all model architectures in Table 15. Consistent with the results for **bach.wav** in Table 2, ReLU-based models struggle to represent high-frequency audio signals. Although optimization with Muon yields substantial improvements for ReLU-based INRs, architectures with periodic activation functions, such as SIREN and FINER, may still provide a more suitable inductive bias for modeling high-frequency audio data with high signal fidelity.

## C.4. Single Image Super-resolution

Due to space constraints, we present representative qualitative results for all model architectures in Figure 11. We notice that Muon-optimized reconstructions tend to have less noisy details. Notably, the Gaussian FFN struggles in both cases, collapsing to a very noisy reconstruction for Adam and the mean color for Muon.

## C.5. CT Reconstruction

Due to space constraints, we present representative qualitative results for all model architectures in Figure 10. Consistent with the quantitative findings, models optimized with Muon show greater relative improvements, particularly for architectures that struggle to capture fine details when trained with Adam, such as ReLU MLP and ReLU FFN. While Muon yields consistent improvements across all models, these gains are less pronounced for architectures like SIREN and FINER, which already achieve high signal fidelity with Adam optimization.

## C.6. Neural Radiance Fields

We present complementary metrics to Table 6 in Table 16, and further quantitative results for four additional scenes from the synthetic NeRF dataset (Mildenhall et al., 2021) in Table 17.

Due to space constraints, we present representative qualitative results for two more scenes from the synthetic NeRF dataset (Mildenhall et al., 2021) in Figure 12 and Figure 13, respectively. While the improvements are less pronounced in the quantitative metrics for NeRFs compared to other tasks, scene reconstructions optimized with Muon feature more details and less noise (e.g., in the drum cymbals).

*Table 16.* **NeRF performance (SSIM & LPIPS)**. Quantitative comparison of NeRF reconstruction quality across different architectures using the Adam and Muon optimizers. We report the mean ± standard deviation over 3 seeds for SSIM and LPIPS. The best-performing optimizer for each architecture is highlighted in **bold**. In a few cases, for scenes marked with †, learning rates from hyperparameter sweeps did not transfer effectively, resulting in non-convergent training, where we resorted to the next stable learning rate in the sweep.

| | SSIM (↑) | | | | | | | |
| --- | --- | --- | --- | --- | --- | --- | --- | --- |
| Methods | Chair | | Drums | | Ficus | | Hotdog | |
| | Adam | Muon | Adam | Muon | Adam | Muon | Adam | Muon |
| ReLU MLP | $0.930 \pm 0.007$ | $\mathbf{0.959 \pm 0.000}$ | $0.786 \pm 0.015$ | $\mathbf{0.860 \pm 0.001}$ | $0.859 \pm 0.007^\dagger$ | $\mathbf{0.912 \pm 0.003}$ | $0.929 \pm 0.011$ | $\mathbf{0.964 \pm 0.002}$ |
| ReLU + PE | $0.972 \pm 0.001$ | $\mathbf{0.974 \pm 0.001}$ | $0.882 \pm 0.002$ | $\mathbf{0.908 \pm 0.001}$ | $0.947 \pm 0.001$ | $\mathbf{0.949 \pm 0.002}$ | $0.959 \pm 0.001$ | $\mathbf{0.964 \pm 0.000}$ |
| InstantNGP | $0.948 \pm 0.000$ | $\mathbf{0.965 \pm 0.001}$ | $0.892 \pm 0.004$ | $\mathbf{0.905 \pm 0.001}$ | $0.934 \pm 0.001$ | $\mathbf{0.941 \pm 0.001}$ | $0.923 \pm 0.009^\dagger$ | $\mathbf{0.955 \pm 0.001}$ |
| SIREN | $0.968 \pm 0.001$ | $\mathbf{0.974 \pm 0.000}$ | $0.904 \pm 0.002$ | $\mathbf{0.908 \pm 0.002}$ | $0.950 \pm 0.001$ | $\mathbf{0.953 \pm 0.001}$ | $0.961 \pm 0.002$ | $\mathbf{0.963 \pm 0.001}$ |
| FINER | $0.961 \pm 0.006$ | $\mathbf{0.974 \pm 0.001}$ | $0.900 \pm 0.004$ | $\mathbf{0.909 \pm 0.003}$ | $0.951 \pm 0.002$ | $\mathbf{0.953 \pm 0.003}$ | $0.953 \pm 0.011$ | $\mathbf{0.962 \pm 0.001}$ |

| | LPIPS (↓) | | | | | | | |
| --- | --- | --- | --- | --- | --- | --- | --- | --- |
| Methods | Chair | | Drums | | Ficus | | Hotdog | |
| | Adam | Muon | Adam | Muon | Adam | Muon | Adam | Muon |
| ReLU MLP | $0.064 \pm 0.004$ | $\mathbf{0.041 \pm 0.000}$ | $0.267 \pm 0.040$ | $\mathbf{0.150 \pm 0.000}$ | $0.193 \pm 0.029^\dagger$ | $\mathbf{0.097 \pm 0.005}$ | $0.055 \pm 0.008$ | $\mathbf{0.021 \pm 0.001}$ |
| ReLU MLP + PE | $0.016 \pm 0.001$ | $\mathbf{0.014 \pm 0.001}$ | $0.089 \pm 0.002$ | $\mathbf{0.057 \pm 0.004}$ | $0.031 \pm 0.001$ | $\mathbf{0.029 \pm 0.001}$ | $0.026 \pm 0.001$ | $\mathbf{0.025 \pm 0.001}$ |
| InstantNGP | $0.034 \pm 0.001$ | $\mathbf{0.023 \pm 0.001}$ | $0.069 \pm 0.003$ | $\mathbf{0.058 \pm 0.000}$ | $0.041 \pm 0.001$ | $\mathbf{0.035 \pm 0.001}$ | $0.061 \pm 0.007^\dagger$ | $\mathbf{0.037 \pm 0.002}$ |
| SIREN | $0.020 \pm 0.000$ | $\mathbf{0.016 \pm 0.001}$ | $0.059 \pm 0.002$ | $\mathbf{0.056 \pm 0.002}$ | $0.031 \pm 0.001$ | $\mathbf{0.028 \pm 0.001}$ | $\mathbf{0.029 \pm 0.004}$ | $0.030 \pm 0.002$ |
| FINER | $0.024 \pm 0.006$ | $\mathbf{0.015 \pm 0.001}$ | $0.060 \pm 0.004$ | $\mathbf{0.054 \pm 0.002}$ | $0.030 \pm 0.002$ | $\mathbf{0.028 \pm 0.002}$ | $0.043 \pm 0.018$ | $\mathbf{0.029 \pm 0.002}$ |

*Table 17.* **NeRF performance for additional scenes**. Quantitative comparison of NeRF reconstruction quality across different architectures using the Adam and Muon optimizers. We report the mean ± standard deviation over 3 seeds for PSNR, SSIM, and LPIPS. The best-performing optimizer for each architecture is highlighted in **bold**. In a few cases, for scenes marked with †, learning rates from hyperparameter sweeps did not transfer effectively, resulting in non-convergent training.

| | PSNR (↑) | | | | | | | |
| --- | --- | --- | --- | --- | --- | --- | --- | --- |
| Methods | Lego | | Materials | | Mic | | Ship | |
| | Adam | Muon | Adam | Muon | Adam | Muon | Adam | Muon |
| ReLU MLP | $22.36 \pm 0.17$ | $\mathbf{26.65 \pm 0.35}$ | $24.32 \pm 0.05^\dagger$ | $\mathbf{26.60 \pm 0.28}$ | $25.18 \pm 3.65^\dagger$ | $\mathbf{25.91 \pm 6.75}^\dagger$ | $19.29 \pm 0.66$ | $\mathbf{20.19 \pm 0.16}$ |
| ReLU MLP + PE | $29.38 \pm 0.16$ | $\mathbf{30.80 \pm 0.05}$ | $26.43 \pm 0.08$ | $\mathbf{26.88 \pm 0.12}$ | $32.79 \pm 0.01$ | $\mathbf{33.35 \pm 0.21}^\dagger$ | $22.84 \pm 0.13$ | $\mathbf{23.27 \pm 0.07}$ |
| InstantNGP | $28.40 \pm 0.19$ | $\mathbf{29.19 \pm 0.06}$ | $23.76 \pm 0.23^\dagger$ | $\mathbf{24.98 \pm 0.02}$ | $30.37 \pm 0.07$ | $\mathbf{31.74 \pm 0.06}$ | $19.89 \pm 0.25$ | $\mathbf{20.15 \pm 0.15}$ |
| SIREN | $29.26 \pm 0.07$ | $\mathbf{29.59 \pm 0.11}$ | $26.72 \pm 0.01$ | $\mathbf{27.25 \pm 0.05}$ | $33.93 \pm 0.09$ | $\mathbf{34.45 \pm 0.04}$ | $22.27 \pm 0.17$ | $\mathbf{22.73 \pm 0.03}$ |
| FINER | $28.96 \pm 0.46$ | $\mathbf{29.76 \pm 0.08}$ | $26.70 \pm 0.11$ | $\mathbf{27.04 \pm 0.12}$ | $33.57 \pm 0.10$ | $\mathbf{34.16 \pm 0.06}$ | $22.34 \pm 0.07$ | $\mathbf{22.70 \pm 0.11}$ |

| | SSIM (↑) | | | | | | | |
| --- | --- | --- | --- | --- | --- | --- | --- | --- |
| Methods | Lego | | Materials | | Mic | | Ship | |
| | Adam | Muon | Adam | Muon | Adam | Muon | Adam | Muon |
| ReLU MLP | $0.826 \pm 0.004$ | $\mathbf{0.896 \pm 0.002}$ | $0.899 \pm 0.002^\dagger$ | $\mathbf{0.927 \pm 0.003}$ | $0.950 \pm 0.022^\dagger$ | $\mathbf{0.954 \pm 0.025}^\dagger$ | $0.697 \pm 0.018$ | $\mathbf{0.729 \pm 0.005}$ |
| ReLU MLP + PE | $0.947 \pm 0.002$ | $\mathbf{0.962 \pm 0.000}$ | $0.924 \pm 0.001$ | $\mathbf{0.934 \pm 0.000}$ | $0.978 \pm 0.000$ | $\mathbf{0.980 \pm 0.001}^\dagger$ | $0.788 \pm 0.005$ | $\mathbf{0.815 \pm 0.002}$ |
| InstantNGP | $0.938 \pm 0.004$ | $\mathbf{0.949 \pm 0.001}$ | $0.885 \pm 0.006^\dagger$ | $\mathbf{0.909 \pm 0.001}$ | $0.972 \pm 0.001$ | $\mathbf{0.979 \pm 0.000}$ | $0.699 \pm 0.015$ | $\mathbf{0.742 \pm 0.010}$ |
| SIREN | $0.941 \pm 0.001$ | $\mathbf{0.945 \pm 0.001}$ | $0.921 \pm 0.000$ | $\mathbf{0.930 \pm 0.001}$ | $0.981 \pm 0.000$ | $\mathbf{0.983 \pm 0.000}$ | $0.783 \pm 0.003$ | $\mathbf{0.803 \pm 0.001}$ |
| FINER | $0.934 \pm 0.007$ | $\mathbf{0.947 \pm 0.001}$ | $0.920 \pm 0.002$ | $\mathbf{0.927 \pm 0.002}$ | $0.980 \pm 0.001$ | $\mathbf{0.982 \pm 0.000}$ | $0.781 \pm 0.003$ | $\mathbf{0.797 \pm 0.001}$ |

| | LPIPS (↓) | | | | | | | |
| --- | --- | --- | --- | --- | --- | --- | --- | --- |
| Methods | Lego | | Materials | | Mic | | Ship | |
| | Adam | Muon | Adam | Muon | Adam | Muon | Adam | Muon |
| ReLU MLP | $0.178 \pm 0.004$ | $\mathbf{0.116 \pm 0.001}$ | $0.070 \pm 0.002^\dagger$ | $\mathbf{0.037 \pm 0.003}$ | $0.082 \pm 0.044^\dagger$ | $\mathbf{0.067 \pm 0.042}^\dagger$ | $0.333 \pm 0.059$ | $\mathbf{0.265 \pm 0.006}$ |
| ReLU MLP + PE | $0.025 \pm 0.001$ | $\mathbf{0.017 \pm 0.000}$ | $0.031 \pm 0.001$ | $\mathbf{0.030 \pm 0.000}$ | $0.013 \pm 0.000$ | $\mathbf{0.010 \pm 0.001}^\dagger$ | $0.112 \pm 0.003$ | $\mathbf{0.092 \pm 0.001}$ |
| InstantNGP | $0.032 \pm 0.002$ | $\mathbf{0.028 \pm 0.001}$ | $0.079 \pm 0.010^\dagger$ | $\mathbf{0.049 \pm 0.000}$ | $0.021 \pm 0.001$ | $\mathbf{0.014 \pm 0.000}$ | $0.229 \pm 0.023$ | $\mathbf{0.164 \pm 0.006}$ |
| SIREN | $0.038 \pm 0.002$ | $\mathbf{0.037 \pm 0.001}$ | $0.036 \pm 0.001$ | $\mathbf{0.032 \pm 0.000}$ | $0.013 \pm 0.000$ | $\mathbf{0.012 \pm 0.000}$ | $0.122 \pm 0.002$ | $\mathbf{0.110 \pm 0.002}$ |
| FINER | $0.039 \pm 0.007$ | $\mathbf{0.030 \pm 0.001}$ | $0.039 \pm 0.002$ | $\mathbf{0.034 \pm 0.001}$ | $0.013 \pm 0.001$ | $\mathbf{0.010 \pm 0.000}$ | $0.119 \pm 0.001$ | $\mathbf{0.111 \pm 0.002}$ |

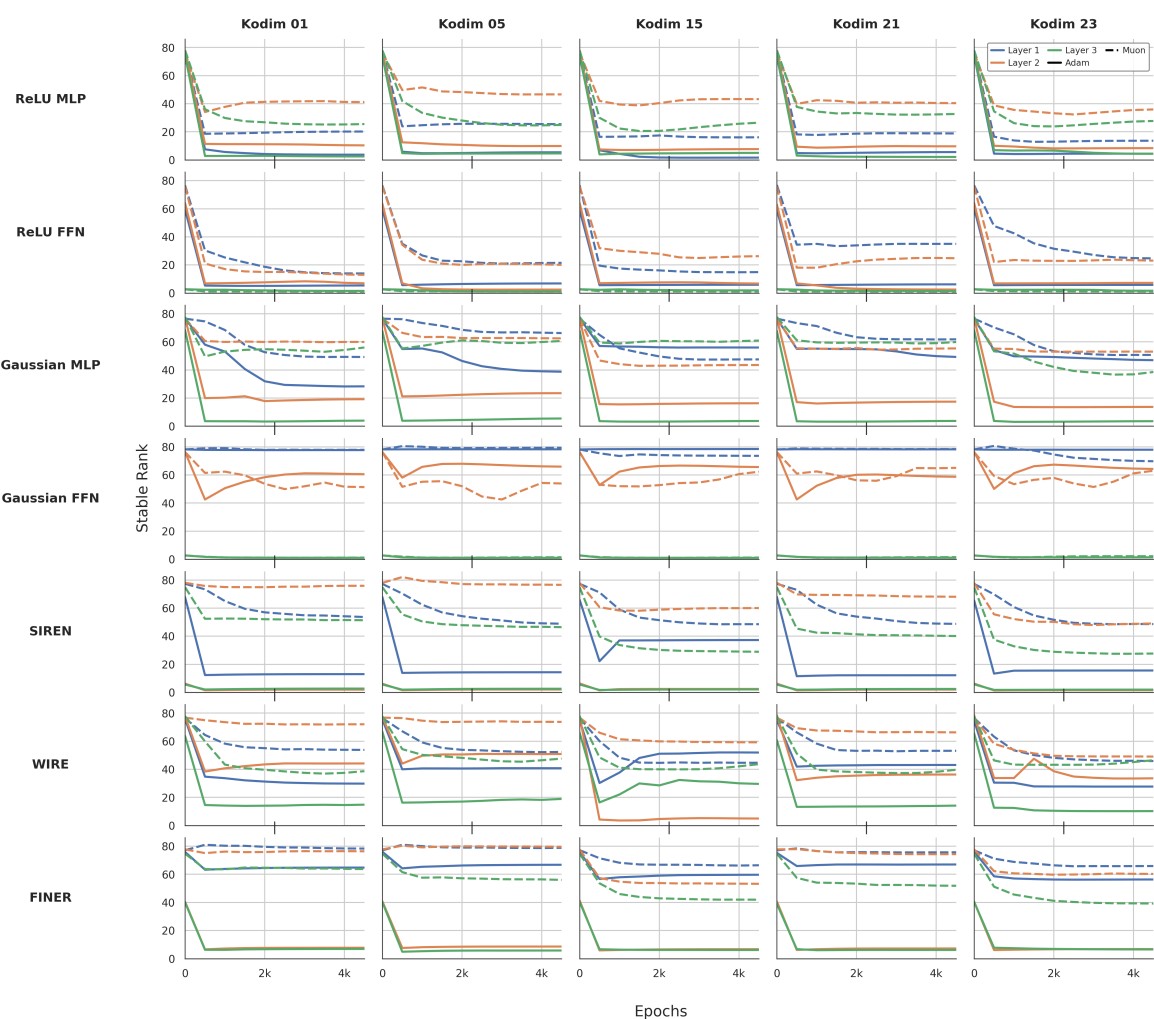

*Figure 6.* Stable Rank plots for all models in the image overfitting experiment.

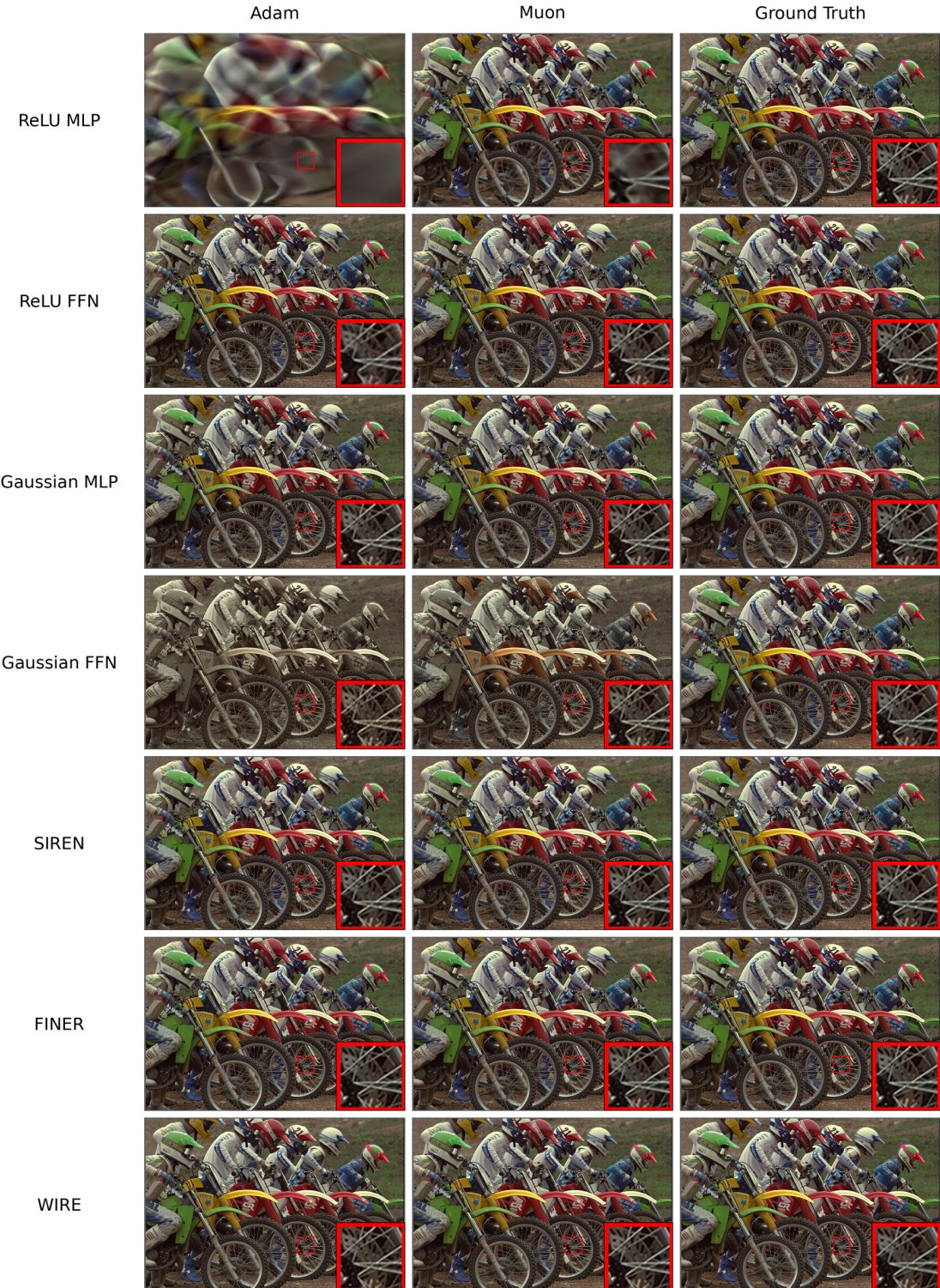

*Figure 7.* Qualitative results for the *kodim05.png* image overfitting experiment.

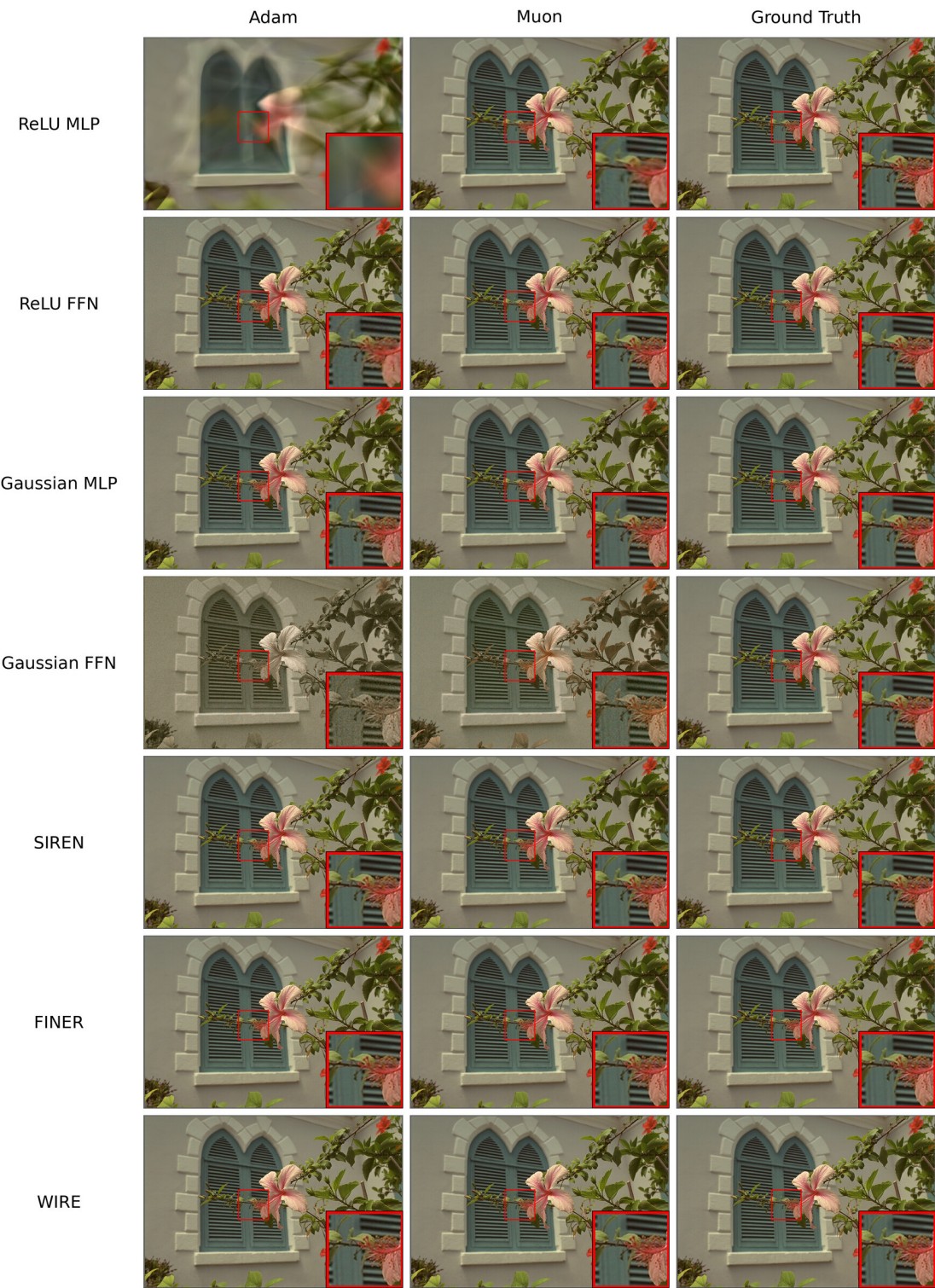

*Figure 8.* Qualitative results for the *kodim07.png* image overfitting experiment.

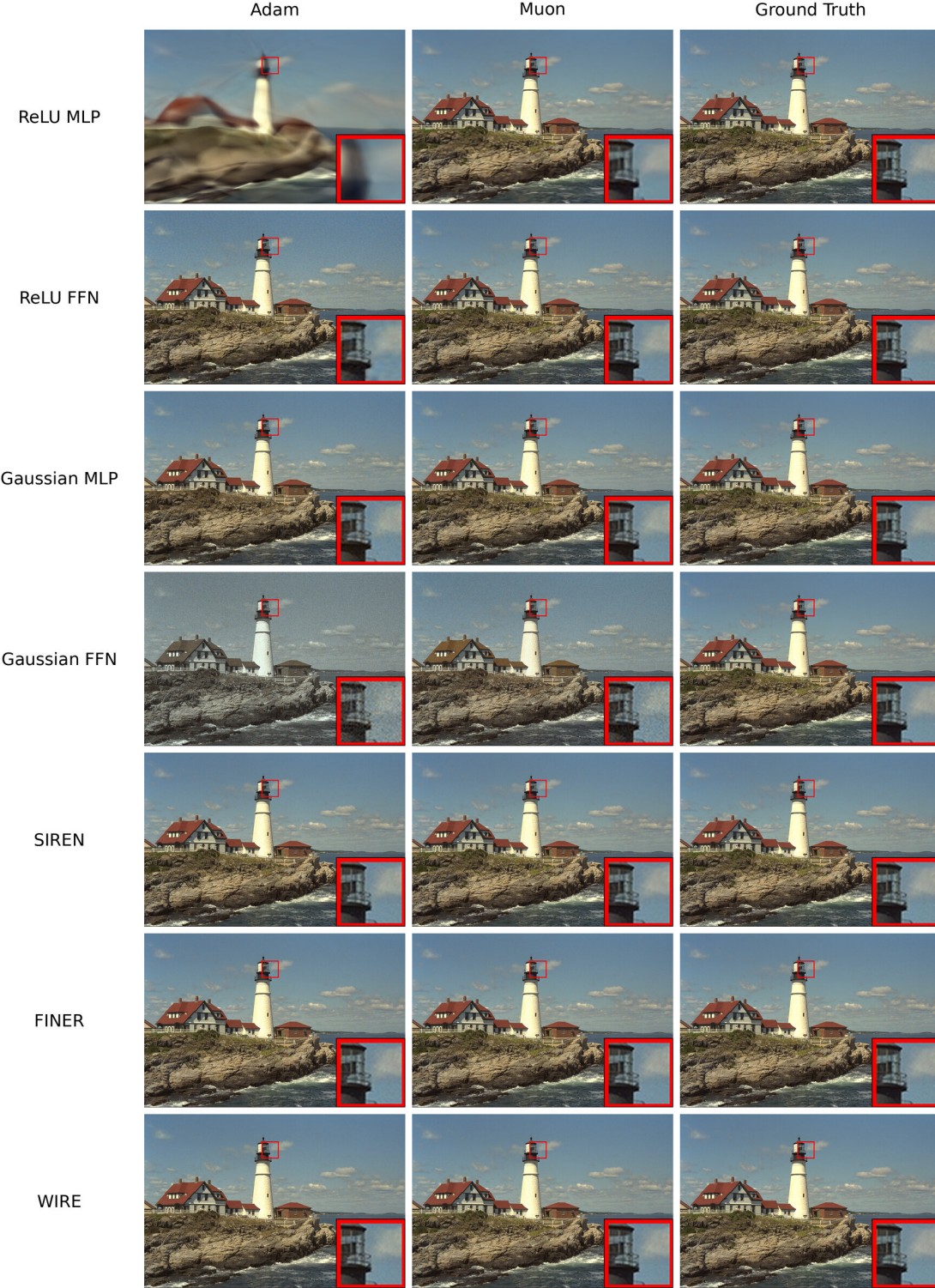

*Figure 9.* Qualitative results for the *kodim21.png* image overfitting experiment.

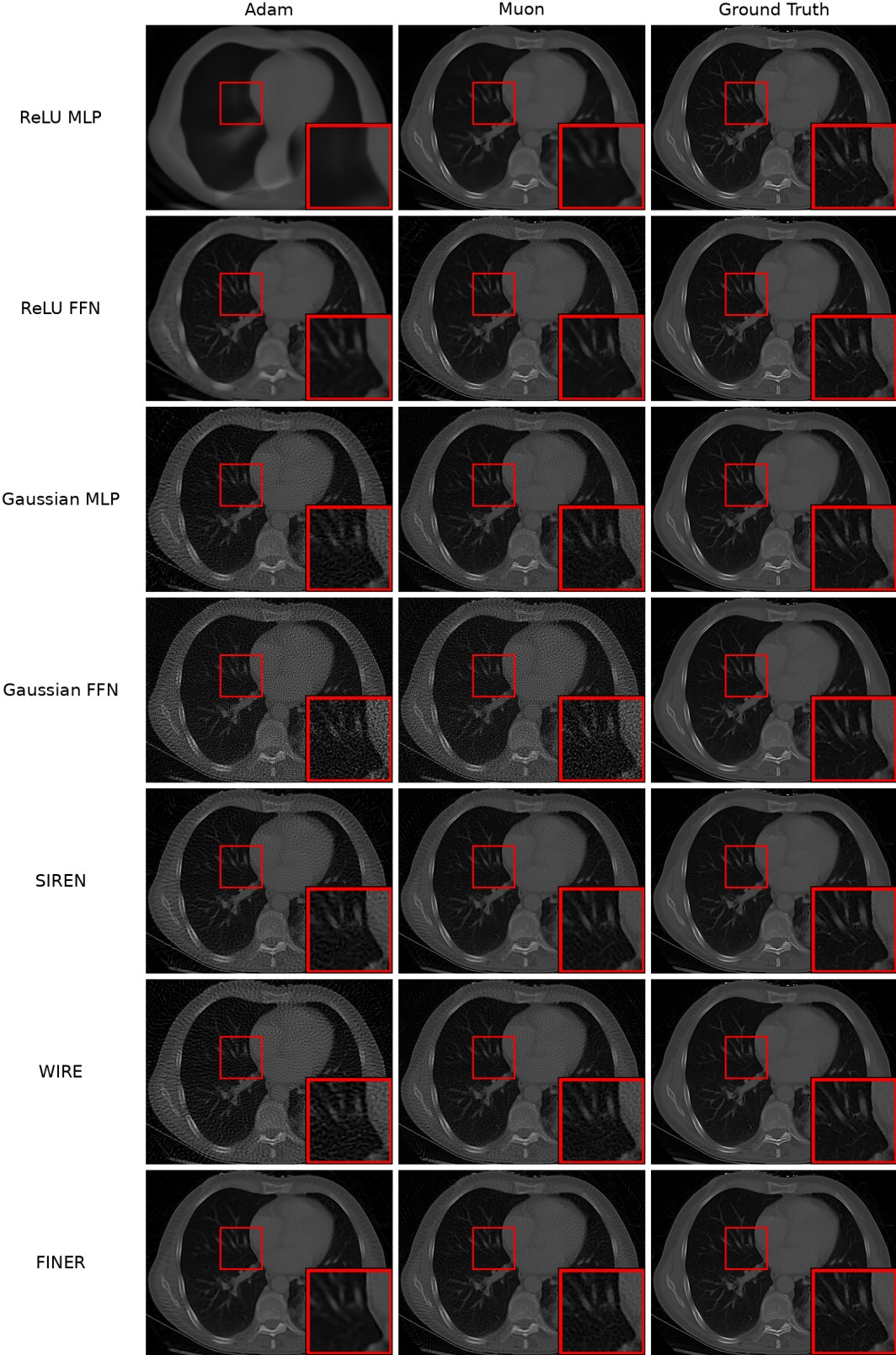

*Figure 10.* Qualitative comparison for the CT reconstruction experiment.

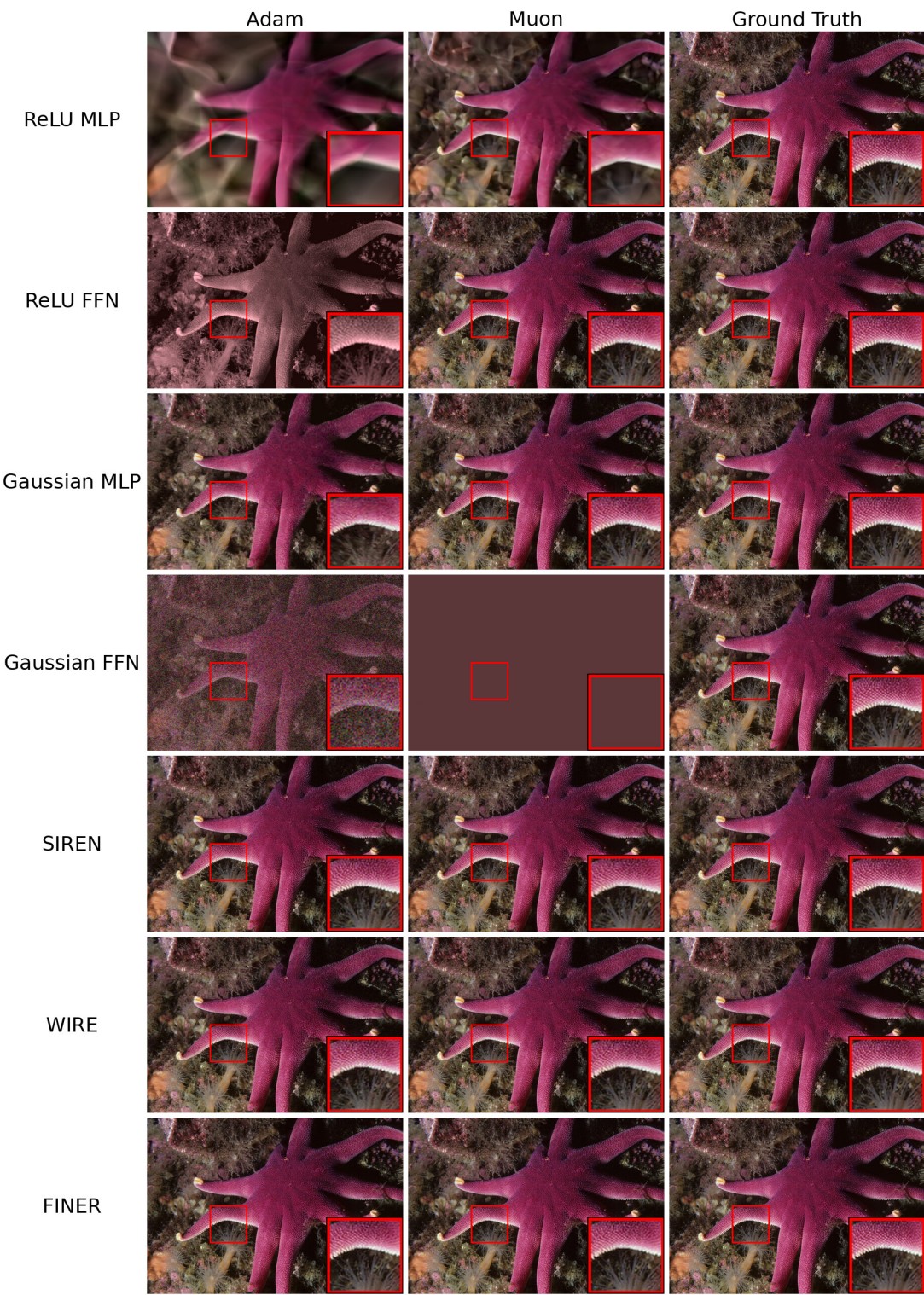

*Figure 11.* Qualitative results for SISIR experiment for *0001.png* from the DIV2K dataset using an upsampling factor of four.

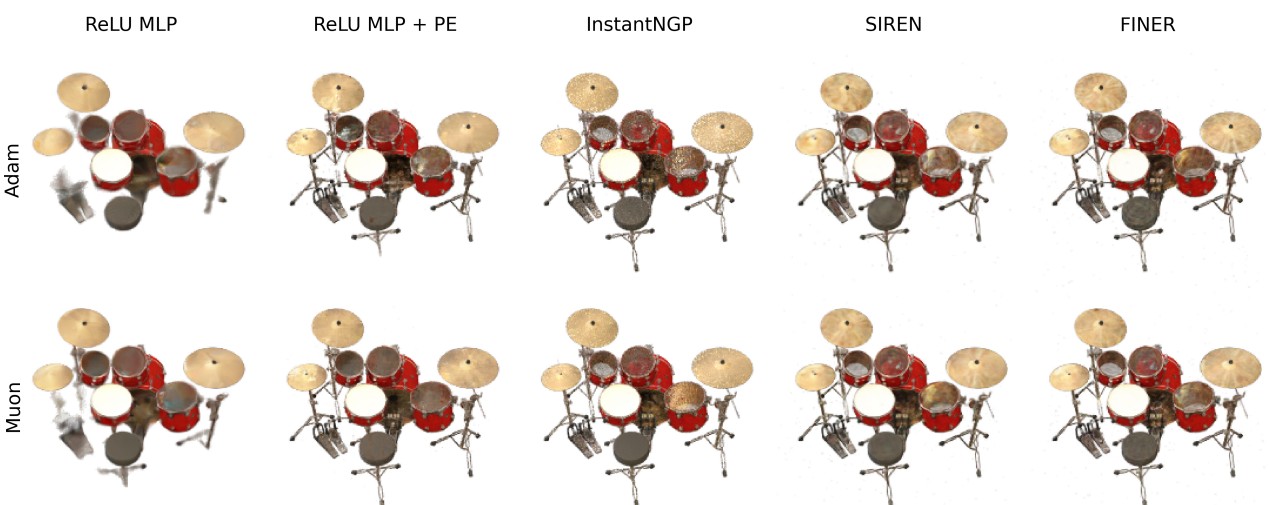

*Figure 12.* Reconstructing a view from a neural radiance field representing a 360-degree scene of a drum set.

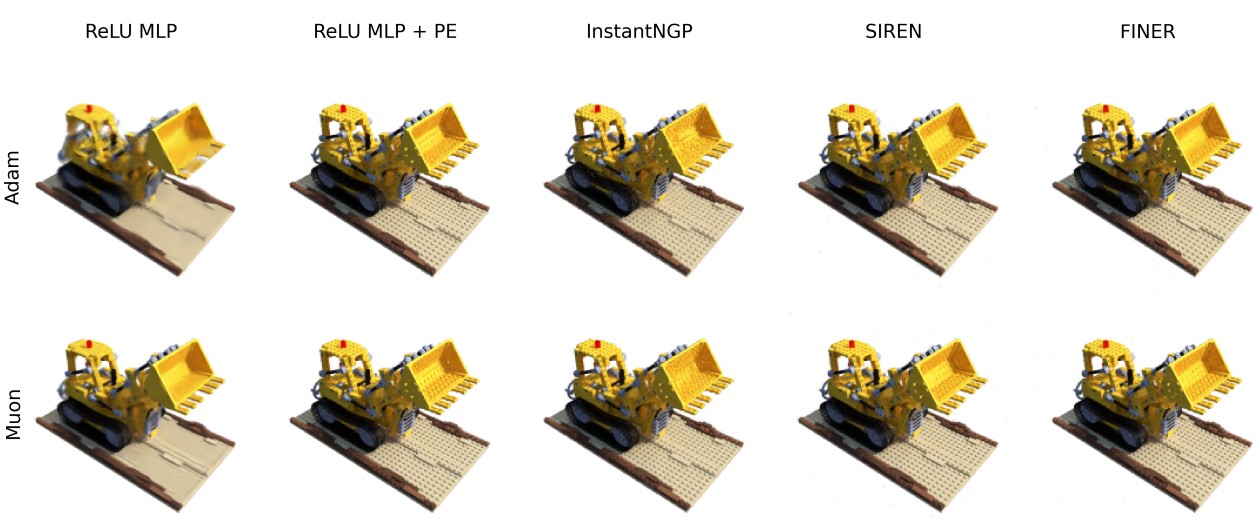

*Figure 13.* Reconstructing a view from a neural radiance field representing a 360-degree scene of a lego excavator.

# D. Ablations

## D.1. Optimizer Ablation

To further study the impact of optimization on stable rank and reconstruction performance, we additionally ablate the established, sign-based optimizer, Lion (Chen et al., 2023b), using the experimental setup described in subsubsection 4.1.1. We similarly conduct a learning rate sweep as detailed in Appendix A.

We report both stable rank and reconstruction performance in Table 18 and Table 19, respectively.

Following experiments with Lion, we see two consistent patterns that emerge:

1. Optimization controls rank independently of architecture. Lion does not consistently preserve stable rank across layers and underperforms Muon in both rank and reconstruction quality. For vanilla ReLU MLPs, Muon mitigates rank collapse and improves PSNR from 22.8 to 30.0 dB.

2. Input- and activation-level interventions reshape where rank is expressed but do not prevent collapse. FFN and SIREN/WIRE/FINER models concentrate high rank in early layers but degrade deeper. Muon preserves high rank throughout; Lion provides only partial improvement.

Thus, a single optimizer change yields up to a +9 dB increase in PSNR without architectural changes or additional parameters.

## D.2. Audio Ablation Study

To further investigate the challenges ReLU MLPs face in representing audio signals, we conduct an ablation study on a shorter temporal signal. We extract a 200ms excerpt from the **bach.wav** audio signal and evaluate reconstruction performance across optimizers.

As shown in Table 20, shortening the audio clip and extending training duration significantly improve reconstruction quality for both optimizers. For context, (Essakine et al., 2025) trains for (only) 2,000 epochs on the full 6-second audio clip in their benchmark paper, which is sufficient for more expressive models to faithfully represent the whole audio clip. When trained for 20,000 epochs, Muon achieves high-fidelity reconstruction with SNR and SI-SNR values exceeding 23 dB, while Adam reaches moderate quality at approximately 5 dB. At 5,000 epochs, the gap widens substantially: Muon demonstrates reasonable reconstruction quality (above 13 dB), whereas Adam completely fails to learn meaningful representations (1 dB). These results demonstrate that ReLU MLPs can effectively represent au-

dio signals, or at least short excerpts, given (1) sufficient training time and (2) suitable optimization.

## D.3. Input Encoding Robustness

While we employ optimal architecture-specific hyperparameters reported in recent studies (Saragadam et al., 2023; Essakine et al., 2025; Kim & Fridovich-Keil, 2025) selected for Adam, it is worthwhile to examine the impact of varying architecture-specific hyperparameters such as $\sigma$ for FFN-based INRs, which interact with the optimization dynamics and change the inductive bias of the network.

Therefore, we conduct a hyperparameter sweep for ReLU-FFNs on kodim01/Section 4.1.1. We report performance and the stable ranks of the hidden layers (SR_1, SR_2) across 5 seeds to connect the results to our theoretical framework.

Table 21 indicates that even when $\sigma$ is tuned, orthogonalized optimization provides substantial (!) gains of up to approx 9 dB. Both optimizers peak at the same sigma values, indicating optimization provides an additive improvement rather than a compensation for suboptimal embeddings.

Notably, while Adam maintains approx. constant rank, Muon dynamically adjusts for suboptimal $\sigma$, suggesting adaptation to the stable rank of the input parameterization, in line with our framework.

## D.4. Grid-based Architectures

Given the recent success of plane-based architectures (Kim & Fridovich-Keil, 2025), we additionally evaluate GA-Planes (Sivgin et al., 2025) on the image reconstruction task of subsubsection 4.1.1 under an identical experimental setup, c.f. subsection B.1. We adopt the architecture of Kim & Fridovich-Keil (2025) employing a 2D feature-grid representation with two axis-aligned line grids and a low-resolution plane grid. All grids are bilinearly interpolated and combined either additively (*Sum*) or multiplicatively (*Multiply*), and decoded by a two-layer ReLU MLP. Since we do not intend to compare GA-Planes to other architectures in this ablation, but rather attempt to isolate the effect of optimization, we adhere to the parameter budget employed in (Kim & Fridovich-Keil, 2025) and re-use their architectural design. We sweep Adam and Muon learning rates as in Appendix B, which confirms the employed learning rates of $lr = 1 \cdot 10^{-2}$ for Adam and, for Muon, a learning rate of $lr = 1 \cdot 10^{-1}$ (Multiply) and $lr = 3 \cdot 10^{-2}$ (Sum) respectively. As Muon updates only the decoder's hidden weight matrix, the remaining parameters, such as the feature grids, bias vectors, and output projection, are optimized by the auxiliary Adam optimizer, for which the sweep selects $lr = 3 \cdot 10^{-2}$ (Multiply) and $lr = 1 \cdot 10^{-2}$ (Sum). We report the results below.

*Table 18.* **Stable rank of hidden layers at convergence**. Stable rank of hidden layers (L1 and L2) at convergence, averaged over 24 Kodak images. The highest stable rank for each architecture and layer type is highlighted in **bold**.

| Model | L1 | | | L2 | | |
|---|---|---|---|---|---|---|
| | Adam | Lion | Muon | Adam | Lion | Muon |
| ReLU MLP | $4.31_{\pm 1.56}$ | $1.46_{\pm 0.11}$ | $\mathbf{18.79}_{\pm 3.29}$ | $8.43_{\pm 1.49}$ | $2.13_{\pm 0.12}$ | $\mathbf{39.18}_{\pm 6.26}$ |
| ReLU FFN | $5.86_{\pm 0.54}$ | $1.06_{\pm 0.03}$ | $\mathbf{34.13}_{\pm 14.51}$ | $6.28_{\pm 1.76}$ | $1.61_{\pm 0.07}$ | $\mathbf{15.90}_{\pm 4.73}$ |
| Gauss MLP | $42.94_{\pm 9.16}$ | $8.53_{\pm 0.97}$ | $\mathbf{64.14}_{\pm 10.42}$ | $17.27_{\pm 3.45}$ | $1.50_{\pm 0.19}$ | $\mathbf{58.90}_{\pm 7.33}$ |
| Gauss FFN | $\mathbf{77.95}_{\pm 0.35}$ | $22.48_{\pm 25.28}$ | $74.92_{\pm 4.29}$ | $\mathbf{61.33}_{\pm 3.36}$ | $1.08_{\pm 0.02}$ | $49.64_{\pm 12.95}$ |
| SIREN | $16.00_{\pm 5.76}$ | $26.35_{\pm 5.61}$ | $\mathbf{44.67}_{\pm 6.72}$ | $1.88_{\pm 0.26}$ | $9.52_{\pm 3.79}$ | $\mathbf{60.69}_{\pm 9.22}$ |
| WIRE | $36.70_{\pm 10.94}$ | $6.84_{\pm 0.87}$ | $\mathbf{50.79}_{\pm 5.43}$ | $29.51_{\pm 17.04}$ | $59.83_{\pm 8.53}$ | $\mathbf{64.50}_{\pm 5.34}$ |
| FINER | $63.33_{\pm 2.68}$ | $35.88_{\pm 10.37}$ | $\mathbf{73.35}_{\pm 4.34}$ | $7.41_{\pm 0.70}$ | $13.70_{\pm 12.27}$ | $\mathbf{70.32}_{\pm 6.59}$ |

*Table 19.* **Image overfitting performance**. Quantitative comparison of reconstruction quality for different architectures using the Adam, Lion, and Muon optimizers. We report the mean and standard deviation for PSNR over 24 Kodak images. The best-performing optimizer for each architecture is highlighted in **bold**.

| Model | PSNR ($\uparrow$) | | |
|---|---|---|---|
| | Adam | Lion | Muon |
| ReLU MLP | $22.83_{\pm 2.85}$ | $24.34_{\pm 3.08}$ | $\mathbf{30.04}_{\pm 3.59}$ |
| ReLU FFN | $31.21_{\pm 2.71}$ | $30.63_{\pm 3.42}$ | $\mathbf{40.50}_{\pm 3.29}$ |
| Gauss MLP | $35.28_{\pm 1.85}$ | $35.25_{\pm 2.73}$ | $\mathbf{41.05}_{\pm 3.06}$ |
| Gauss FFN | $25.20_{\pm 2.18}$ | $24.69_{\pm 2.12}$ | $\mathbf{28.13}_{\pm 2.08}$ |
| SIREN | $39.33_{\pm 2.64}$ | $41.75_{\pm 2.96}$ | $\mathbf{42.33}_{\pm 3.29}$ |
| WIRE | $32.43_{\pm 9.56}$ | $39.73_{\pm 3.36}$ | $\mathbf{42.99}_{\pm 3.34}$ |
| FINER | $39.87_{\pm 2.04}$ | $42.46_{\pm 3.40}$ | $\mathbf{42.59}_{\pm 2.74}$ |

*Table 20.* Reconstruction performance of ReLU MLPs for representing a 200ms audio excerpt (*middle_200ms_clip.wav*). Mean $\pm$ std over 5 seeds.

| Optimizer | SNR (dB) | SI-SNR (dB) |
|---|---|---|
| Adam (5000 epochs) | $1.83_{\pm 0.55}$ | $-2.92_{\pm 1.46}$ |
| Muon (5000 epochs) | $13.13_{\pm 0.31}$ | $12.92_{\pm 0.32}$ |
| Adam (20000 epochs) | $4.91_{\pm 0.54}$ | $3.22_{\pm 0.80}$ |
| Muon (20000 epochs) | $23.51_{\pm 0.98}$ | $23.49_{\pm 0.98}$ |

### D.4.1. GA-PLANES

In line with the main results' key observations, we note that stable rank decay is also pronounced for architectures such as GA-Planes, which feature a large feature projection stage and a comparatively small decoder. Importantly, Muon elevates the stable rank of the small decoder, inducing richer representations and expressiveness in the decoder, attaining higher reconstruction quality for both stages.

Specifically, under standard Adam optimization, GA-Planes exhibits a severely collapsed stable rank (in the range of 5-6). Conversely, orthogonalized optimization via Muon yields an approximately four-fold increase in stable rank, irrespective of the feature combination technique. By preventing this collapse, Muon induces richer representations in the decoder, resulting in sharper reconstructions and gains of up to 2 dB in PSNR.

As detailed in Table 22, these results validate three core predictions of the stable rank framework:

1. **Feature combination sets the rank ceiling:** The choice of combination operation fundamentally limits the maximum achievable rank.

2. **Additive constraints:** Summation imposes a low rank ceiling.

3. **Multiplicative capacity:** The Hadamard product provides a higher ceiling, which mathematically explains the significant +3.53 dB gap between Multiply and Sum under Adam.

This ablation experiment reveals that the impact of optimization is more pronounced under tighter architectural bottlenecks. As shown in Table 22, Muon achieves a larger relative gain for summation than for multiplication. This may be intuitively explained by the lower rank ceiling, which leaves more room for recovery. Finally, this nicely illustrates that input-level and optimization-level interventions are complementary axes, and combining both (Multiply + Muon) achieves the highest reconstruction performance.

### D.4.2. DECODER CAPACITY

We ablate the effect of decoder capacity by varying the hidden-layer width of the decoder MLP for the same image reconstruction experiment subsubsection 4.1.1. Starting from the GA-Planes (Sum) representation trained with Muon, we sweep the decoder hidden width over $32, 64, 128$ while holding the feature-grid encoder, the feature dimension, and the learning rates fixed. Given the comparatively small footprint of the decoder MLP changes in comparison to the feature grids, the total parameter count stays within a narrow range. All configurations are evaluated on the full dataset.

As shown in Table 23, widening the decoder monotonically increases both the stable rank of the decoder's hidden layer ($7.77 \rightarrow 12.94 \rightarrow 22.48$) and the reconstruction quality ($28.14 \rightarrow 30.74 \rightarrow 34.15$ dB). Decoder capacity is therefore a direct lever on the attainable stable rank, mirroring

*Table 21.* **Effect of $\sigma$ on performance and stable rank**. Quantitative comparison of PSNR and hidden layer stable rank (SR$_1$, SR$_2$) using the Adam versus Muon optimizers across different values of $\sigma$. The $\Delta$ column indicates the PSNR improvement of Muon over Adam. The highest values for PSNR and stable rank are highlighted in **bold**.

| $\sigma$ | PSNR ($\uparrow$) | | $\Delta$ | SR$_1$ | | SR$_2$ | |
| --- | --- | --- | --- | --- | --- | --- | --- |
| | Adam | Muon | | Adam | Muon | Adam | Muon |
| 1 | $23.32_{\pm 0.38}$ | $\mathbf{24.70}_{\pm 1.84}$ | $+1.38$ | $8.2_{\pm 1.4}$ | $\mathbf{47.4}_{\pm 3.5}$ | $5.3_{\pm 1.2}$ | $\mathbf{25.8}_{\pm 3.6}$ |
| 2 | $25.11_{\pm 0.21}$ | $\mathbf{29.90}_{\pm 0.43}$ | $+4.79$ | $6.7_{\pm 0.5}$ | $\mathbf{56.1}_{\pm 2.6}$ | $5.3_{\pm 1.7}$ | $\mathbf{27.8}_{\pm 2.0}$ |
| 5 | $27.11_{\pm 0.16}$ | $\mathbf{36.06}_{\pm 0.80}$ | $+8.94$ | $6.1_{\pm 0.5}$ | $\mathbf{15.4}_{\pm 4.0}$ | $5.6_{\pm 1.1}$ | $\mathbf{14.3}_{\pm 3.4}$ |
| 10 | $27.61_{\pm 0.20}$ | $\mathbf{36.15}_{\pm 0.10}$ | $+8.54$ | $5.5_{\pm 0.2}$ | $\mathbf{13.7}_{\pm 1.1}$ | $5.8_{\pm 1.2}$ | $\mathbf{11.6}_{\pm 1.8}$ |
| 20 | $27.56_{\pm 0.25}$ | $\mathbf{34.77}_{\pm 0.19}$ | $+7.21$ | $5.9_{\pm 0.5}$ | $\mathbf{31.2}_{\pm 3.4}$ | $5.2_{\pm 1.4}$ | $\mathbf{8.3}_{\pm 2.7}$ |
| 50 | $22.98_{\pm 0.53}$ | $\mathbf{29.06}_{\pm 1.31}$ | $+6.08$ | $6.8_{\pm 0.5}$ | $\mathbf{22.1}_{\pm 4.2}$ | $2.7_{\pm 0.7}$ | $\mathbf{21.4}_{\pm 5.8}$ |

*Table 22.* Stable rank of the last hidden decoder layer for GA-Planes and reconstruction quality on the Kodak dataset (24 images, mean $\pm$ standard deviation). Muon greatly elevates the decoder's stable rank for GA-Planes relative to Adam, where rank decay is pronounced. We include ReLU-MLP values for reference to illustrate that observations are consistent across decoding stages of a ReLU-MLP INR.

| Method | Feature Combination | Optimizer | Stable Rank $\uparrow$ | PSNR (dB) $\uparrow$ |
| --- | --- | --- | --- | --- |
| ReLU MLP | – | Adam | $3.82_{\pm 1.39}$ | $22.83_{\pm 2.85}$ |
| | | Muon | $\mathbf{28.29}_{\pm 3.04}$ | $\mathbf{30.04}_{\pm 3.59}$ |
| GA-Planes | Sum | Adam | $5.75_{\pm 0.66}$ | $33.66_{\pm 2.55}$ |
| | | Muon | $27.24_{\pm 3.17}$ | $\mathbf{35.70}_{\pm 3.24}$ |
| | Multiply | Adam | $5.24_{\pm 0.67}$ | $37.19_{\pm 3.00}$ |
| | | Muon | $20.16_{\pm 3.39}$ | $\mathbf{38.57}_{\pm 3.26}$ |

*Table 23.* GA-Planes (Sum, Muon) decoder hidden-dimension ablation on the Kodak dataset (24 images, mean $\pm$ standard deviation, seed 42). Stable rank is measured at the decoder hidden layer; widening the decoder raises both the hidden-layer stable rank and reconstruction quality.

| Hidden Dim | Stable Rank $\uparrow$ | PSNR (dB) $\uparrow$ |
| --- | --- | --- |
| 32 | $7.77_{\pm 1.18}$ | $28.14_{\pm 3.04}$ |
| 64 | $12.94_{\pm 2.21}$ | $30.74_{\pm 3.17}$ |
| 128 | $22.48_{\pm 4.13}$ | $34.15_{\pm 3.35}$ |

the increase in rank observed when switching from Adam to Muon.

Mathematically, this is demonstrated by the gradient equation $\nabla_{W_l}\mathcal{L} = G_{l+1}H_l^\top$, which implies that the rank of the gradient is restricted such that $\mathrm{rank}(\nabla_{W_l}\mathcal{L}) \leq \min(\mathrm{rank}(F), h)$.

Ultimately, while the decoder width $h$ dictates the maximum possible rank and the feature combination strategy manages rank growth, the decoder of high-rank input architectures, such as GA-Planes (Sivgin et al., 2025) or HashGrid (Müller et al., 2022), still remains highly susceptible to collapse during optimization unless explicitly preserved.

