# OpenReview forum: "Optimizing Rank for High-Fidelity Implicit Neural Representations"
_ICML.cc/2026/Conference — ICML 2026 regular_

### Official Review · Reviewer_Cui8 · 2026-02-27

**Soundness:** 3
**Presentation:** 3
**Significance:** 4
**Originality:** 3
**Overall Recommendation:** 5
**Confidence:** 4

**Summary:**

The authors highlight the wider belief that overcoming spectral bias (inability of MLPs to learn high-frequency signals efficiently) is a major challenge for MLP-based implicit neural representations (INRs). Authors claim, however, spectral bias is not an inherent property of MLPs, rather being caused by rank degradation during training. This rank degradation is measured using the ‘stable rank’ metric. Authors propose to address this degradation not through architectural changes, but by using the Muon optimizer, and proceed to extensively test this approach, showing a significant performance improvement across multiple architectures and tasks.

**Compliance With Llm Reviewing Policy:**

Affirmed.

**Final Justification:**

Authors clarified my concerns and are committed to improving the paper and extending it with additional ablation studies.

**Key Questions For Authors:**

I asked my key quesiton in section B.2.

I would also be happy to hear more about the intuition of authors on spectral bias, but it won't affect my score.

**Limitations:**

Yes.

**Strengths And Weaknesses:**

**The paper is clearly written, with a good discussion of related works and their focus on architectural improvements and analysis of INRs through the lens of Fourier Transform/NTK. The motivation for using stable rank and its relationship with network expressiveness is also well explained (in section 3.1).**

I agree that selection of an optimizer for INR training is an underappreciated topic and work in this area can have a high impact on future research. This is especially true as selection of a good optimizer is a choice largely orthogonal to previous methods for increasing INR performance, and can be applied independently. The wide applicability of the training strategy proposed in the paper is experimentally verified on a range of architectures (Siren, Finer, Wire, etc.) and application domains (image, audio, SDF, object reconstruction), which provides experimental proof for the validity of proposed training strategy. Additionally, hyperparameters used in experiments are provided in the appendix, and code will be released, making this work reproducible.

I believe this is valuable work that will shape further INR research, nevertheless, I have a few comments.

## A. Spectral bias
My biggest concern with this work is the relatively small weight placed on spectral bias (the inability of networks to learn high frequency signals using a reasonable compute budget). The typical solution to this problem in INRs is based on embedding layers and has a significant impact on performance - many existing models can improve performance just by reconfiguring this single layer [1]. As such, comparing different architectures without adjusting frequency of the embedding layer (e.g. by grid search, or methods such as FreSh [1]) is not very convincing (e.g. claim from lines 260-262 - discussed below). This is partially noted in lines 165-167, but a broader discussion would be beneficial.

Let me specifically highlight the following parts of the paper:
1. Lines 260-262: **the authors claim a performance gap between architectures was closed when applying Muon, but the hyperparameters of the embedding layers doesn’t seem to have been adjusted**. As discussed above, configuration of this layer is crucial for performance, making this claim relatively weak - one could similarly "close" the performance gap by configuring the embedding layer poorly configured (e.g. scale of 0.1 for Fourier Features). Additionally, the gap is still present if Muon is applied equally to both methods, not only to the worse-performing architecture.


1. Lines 169-171: the authors claim that input-level interventions fail to overcome spectral bias. Could authors expand on this claim? My understanding is that effective learning rate is low for high frequencies (as discussed in [2]), but input-level mappings allow an INR to learn a low frequency signal instead (for which the effective learning rate is high), thus effectively solving the spectral bias issue and contradicting this claim.



## B. Experiments

While the results are broad and convincing, I believe they could benefit from a few improvements:
In lines 202-210, a pre-training approach is presented, together with a claim that it increases the rank. However, **experimental verification in Figure 2 focuses on model performance, not showing the rank increase**. I would like to ask authors to include this information in Figure 2, or in the Appendix. Additionally, the differences in the top row of Figure 2 are relatively small, so I would also suggest including performance (e.g. PSNR) of each approach.

Currently, only Table 1 reports standard deviation. **It would strengthen the evaluation if standard deviation was included for all experiments**. I suspect this was skipped due to the small number of datapoints used in particular experiments (e.g. 2 audio samples used for table 2), but in such cases experiments can be repeated over multiple seeds. Additionally, to make statistical significance of the results easier to judge, I would recommend using standard error or confidence intervals instead of standard deviation (e.g. in Table 1).




## C. Other
1. Equation in line 134 doesn’t include bias / it’s not stated bias is a part of W.

1. Sentence in lines 173-175 (left column) is not supported and contradicted by lines 179-181.

[1] Kania, Adam, et al. "FreSh: Frequency Shifting for Accelerated Neural Representation Learning." arXiv, 7 Oct. 2024, doi:10.48550/arXiv.2410.05050.
[2] Basri, Ronen, et al. "The Convergence Rate of Neural Networks for Learned Functions of Different Frequencies." arXiv, 2 June 2019, doi:10.48550/arXiv.1906.00425.

---

> ### Author Rebuttal · Authors · 2026-03-31
>
> We thank you for the positive assessment and detailed engagement. We respond point-by-point below.
>
> > [...] the authors claim a performance gap between architectures was closed [...] but the hyperparameters of the embedding layers doesn’t seem to have been adjusted.
>
> > [...] comparing different architectures without adjusting frequency of the embedding layer [...]
>
> We agree adjusting architecture-specific parameters, such as $\sigma$ for FFNs, can "make or break" reconstruction. To ensure fair experiments, we followed parameters that were found to be optimal for these tasks in literature [1,2].
>
> However, you raise an important question: _do optimizer-level gains persist when input-level hyperparameters are also tuned?_
>
> To answer this, we conduct a hyperparameter sweep for ReLU-FFNs on kodim01/Section 4.1.1. We report performance together with stable rank of the hidden layers (SR₁, SR₂) across 5 seeds to connect the results to our theoretical framework.
>
> |σ|AdamPSNR↑|MuonPSNR↑|Δ|SR₁(Adam)|SR₁(Muon)|SR₂(Adam)|SR₂(Muon)|
> |---|---|---|---|---|---|---|---|
> |1|23.32±0.38|24.70±1.84|+1.38|8.2±1.4|47.4±3.5|5.3±1.2|25.8±3.6|
> |2|25.11±0.21|29.90±0.43|+4.79|6.7±0.5|56.1±2.6|5.3±1.7|27.8±2.0|
> |5|27.11±0.16|36.06±0.80|+8.94|6.1±0.5|15.4±4.0|5.6±1.1|14.3±3.4|
> |10|27.61±0.20|36.15±0.10|+8.54|5.5±0.2|13.7±1.1|5.8±1.2|11.6±1.8|
> |20|27.56±0.25|34.77±0.19|+7.21|5.9±0.5|31.2±3.4|5.2±1.4|8.3±2.7|
> |50|22.98±0.53|29.06±1.31|+6.08|6.8±0.5|22.1±4.2|2.7±0.7|21.4±5.8|
>
> *Results directly address concern*: A hyperparameter sweep for ReLU-FFNs on kodim01/Section 4.1.1 shows that even when $\sigma$ is tuned, orthogonalized optimization provides substantial (!) gains of up to approx 9 dB. Both optimizers peak at the same sigma values, indicating optimization provides an additive improvement rather than a compensation for suboptimal embeddings.
>
> Notably, while Adam maintains approx. constant rank, Muon dynamically adjusts for suboptimal $\sigma$, suggesting adaptation to the stable rank of the input parameterization, in line with our framework.
>
> > [...] one could similarly "close" the performance gap by configuring the embedding layer poorly configured (e.g. scale of 0.1 for Fourier Features).
>
> We do not claim that ReLU MLPs outperform networks with Fourier Features in general. Rather, we show that ReLU MLPs,  traditionally considered less expressive, can substantially close this gap through stable rank-aware optimization, even with carefully tuned Fourier Feature hyperparameters (as introduced above).
>
> Importantly, this strategy also transfers to more expressive architectures, providing consistent gains beyond what is achievable by input-level or embedding-level interventions alone. We view this as evidence for the value of our unified framework. For example, FINER, the best-performing architecture in Table 1, achieves the strongest results by combining activation-level and optimization-level interventions, two complementary axes identified by our stable-rank perspective. The greatest gains thus come from jointly combining these axes.
>
> > [...] results are broad and convincing, I believe they could benefit from a few improvements [...] include this information in Figure 2, or in the Appendix.
>
> Thanks, we will update Figure 2 following your suggestion in the revised manuscript.
>
> >Currently, only Table 1 reports standard deviation. [...] but in such cases experiments can be repeated over multiple seeds.
>
> We stored all results in Wandb CSV files (released upon acceptance), and will update tables based on your recommendation.
> We provide a 5-seed ablation for "Bach" with ReLU FFNs. Low standard deviations confirm that the reported results are robust to random initialization, and quantitative comparison between methods remains valid.
>
> |Optimizer|SI-SNR(dB)|
> |---|---|
> |Adam|7.81±0.55|
> |Muon|19.66±0.34|
>
> > Equation in line 134 doesn’t include bias / it’s not stated bias is a part of W.
>
> We will revise and tighten Eq. 1 to include the bias term:
> $$
> H^{l+1} = W^l H^l + b^l
> $$
> The subsequent gradient factorization remains unchanged, as the bias does not affect the subspace structure.
>
> > Sentence in lines 173-175 (left column) is not supported and contradicted by lines 179-181.
>
> We agree that original wording overstates the implication of Eq. 2. While Eq. (2) establishes stable rank of the update is upper-bounded by the rank of the activations, it does not imply the update is fully determined by them. We will revise accordingly.
>
> We invite you to examine further ablations on optimization (F9g6) and GAPlanes (RegR). We thank you for your great feedback, we are looking forward to the discussion period.
>
> ---
> [1] Kim, Namhoon, and Sara Fridovich-Keil. "Grids Often Outperform Implicit Neural Representation at Compressing Dense Signals." NeurIPS 2025
>
> [2] Essakine, Amer, et al. "Where do we stand with implicit neural representations? a technical and performance survey." TMLR 2024

---

> > ### Author Rebuttal · Reviewer_Cui8 · 2026-04-02
> >
> > Thank you for the additional experiments and clarifications. Will this ablation (and ones from replies to other viewers) be included in the camera-ready manuscript?

---

> > > ### Author Response · Authors · 2026-04-02
> > >
> > > Thank you for the quick reply and positive feedback. Yes, absolutely! We believe the rebuttal phase was very constructive and has helped us to sharpen the framing of our paper, particularly in highlighting the novelty of viewing INRs through a stable rank perspective structured along the three axes of input-level, activation-level and the previously overlooked optimization-level that we introduce. Adding the ablations to the manuscript will help us to improve it further, and we are committed to adding them and discussing all other aspects of the rebuttal in the camera-ready version.

---

### Official Review · Reviewer_zca9 · 2026-03-07

**Soundness:** 1
**Presentation:** 3
**Significance:** 1
**Originality:** 3
**Overall Recommendation:** 2
**Confidence:** 5

**Summary:**

This paper studies the spectral bias of implicit neural representations through the lens of the stable rank. It is has been observed via a variety of works on INRs that the spectral bias they admit during training is due to their architecture. However, the authors challenge this viewpoint claiming that the key driver of spectral bias during training is the degradation of the stable rank of the weights of the network. Motivated by this the authors introduce a new way of training INRs that involve an optimizer intervention on the stable rank that seeks to push the parameters to near-orthogonal updates as for example carried out in optimizers such as Muon. They show that optimizing INRs in this way using simple ReLU based INRs, that are known to suffer from spectral bias and thus often can only fit low frequency detail in a signal, can overcome spectral bias and yield much better performance on a variety of well known INR tasks.

**Compliance With Llm Reviewing Policy:**

Affirmed.

**Final Justification:**

I thank the authors for their rebuttal and discussion. I have chosen to keep my score as reject as I still don't believe the paper is at the level of an ICML publication and that there is any real novelty and contribution.

During the review I pointed out to the authors that they seem to suggest in the paper that they solve a theoretical gap and that this is completely overplayed. The authors responded by saying that they disagree and that they have theoretical contributions given by equations (1) and (2). However, these equation are extremely simple to derive and cannot be taken as a contribution for a publication at the level of ICML. Deriving such equations requires no real novelty. After I responded to the authors' rebuttal pointing this out they responded saying that they submitted their work to the applications in computer vision track implying that they shouldn't be held accountable for the lack of theory. Ordinarily I have no issue with this however the paper claims to make theoretical contributions and solve a theoretical gap, and their rebuttal claimed they make concrete theoretical contributions so if this is the case than I have to look at the novelty of those claims which I did and found none. In terms of the field of INRs and applications they have not really produced any novel application as well. As mentioned in my review, they simply suggest, and give some very simple insight, that stable rank is a factor at play for spectral bias and that one should use this insight to use different optimizers on INRs. They then use Muon to show that they can get better results. However, there is no real contribution as Muon is a known optimizer, all the INRs they use in their experiments are INRs from the literature. This then leads to the question of what the real contribution and novelty of this paper is? And I cannot find anything at the level of an ICML publication.

Giving the authors the benefit of the doubt, I went into the references they cite such as Beyond Periodicity by Ramasinghe et al., WIRE by Sargadam et al., and Curvature aware training by Saratchandran et al. In the first two references I noticed there was clear contribution by way of defining a new activation for INRs and showing how and why they perform much better than previous ones. For the last reference, although the authors use an already known optimizer L-BFGS, the authors prove clear theorems showing that optimizing certain INRs with such optimizers yields super-linear convergence however with optimizers such as SGD one can only get linear convergence. There is clear novelty in these papers the authors cite. My suggestion to the authors would be to read such papers and see how they can enhance their insights at such a level. This would make the contribution much more clear and show real novelty.

As it stands I cannot accept the paper.

**Key Questions For Authors:**

1. What is the real contribution of you work? Is it to identify degrading stable rank as the primary driver of spectral bias?

2. I noticed that while you have a impact statement you don't really have any limitations of the actual method. I would take this to be limitations of applying Muon. Is there any overhead compared to Adam or does using Newton-Schulz to approximate near orthogonal updates make the method very efficient?

3. ReLU MLP when trained with SGD suffers from spectral bias which is the original statement of spectral bias. But when INRs are trained with Adam they do better than SGD thus this already shows that Adam is doing some sort of gradient regularization so does this not mean Muon is just another level of this? It would have been good had you put SGD in the experiments too but I believe on tasks like NeRF it is impossible to train an INR with SGD.

4. Your main contribution to me is that you seem to identify spectral bias as originating due to stable rank degradation. However, I find that adding Muon to just a ReLU MLP does increase performance but it does not seem to do better than say SIREN with Muon on some tasks and even SIREN with Adam in some cases (for example in image overfitting). Doesn't this mean that while Muon is helping ReLU MLP overcome spectral bias it is not doing it as efficiently as say SIREN?

**Limitations:**

The authors did have a Impact Statement at the end of their work. However, they didn't have any limitations about their insights and the method of using Muon which I think would have led to better understanding of how effective Muon is.

**Strengths And Weaknesses:**

**Strengths.**

1. **Presentation:** The presentation of the paper is in general very good. The problem formulation is well stated in the abstract and intro itself and relevant citations are clearly given in the right context in the related work section. The experiments carried out in the experiments section, section 4, are clearly explained and are easy for the reader to get the message the authors are trying to get across.

2. **Originality:** The approach the authors take to study the spectral bias of INRs is novel and original and I haven't really seen an approach to overcome spectral bias via optimizers done before within this context. Furthermore, many of the works in INRs that tackle problems such as spectral bias do so from the viewpoint of the architecture itself so this work does add some value to the INR community. One thing I would say is that when I Googled several of the citations of the authors to check and make sure they were legitimate citations. I found the work "Preconditioners for the Stochastic Training of Neural Fields" in CVPR 2025 by Chng et al. That does take an optimization viewpoint but in a different sense in that they show how to train INRs stochastically with quasi-second order style methods provided the right activation function for the INR is chosen. While they do not look at spectral bias their viewpoint can be seen as saying that the spectral bias of ReLU MLPs is coming from a poorly regularized loss landscape, which they show by computing curvature. This complements your work where you show that the right optimizer can regularize the stable rank which I would hypothesize regularizes the gradient trajectory in some way.

**Weaknesses.**

1. **Significance:** The major issue I feel with this paper is the significance of the contribution. While the authors have highlighted a nice viewpoint on spectral bias of INRs their contribution falls short of a publication at the level of ICML. Their solution is to make sure the gradient updates are near orthogonal which is a good observation but then their methodology is to actually use an optimizer that does this through an approximation, via Newton-Schulz, namely the Muon optimizer. So the real contribution of the paper is more an observation than an actual method. This in itself is not a problem if the authors then showed some theoretical contributions that near-orthogonal updates actually do increase stable rank across training but the authors only show this empirically. Furthermore, I noticed that while the authors say that near orthogonal updates yield better stable rank and this can overcome spectral bias I notice that in some experiments their use of Muon on ReLU MLP still has much lower performance than some of the other activation such as SIREN with Muon. So doesn't this mean that the previous works that apply architectural modifications are still the best way to address spectral bias because the difference between say SIREN with Adam and then with Muon on Image Overfitting is about 2 PSNR but with ReLU MLP it's about 8 PSNR and ReLU MLP with muon still underperforms, by about 9 PSNR, a SIREN with Adam? I notice similar behaviour with say NeRF and for SDFs you don't seem to consider other activations. So it seems to me that the best method would be to still use an activation like SIREN but also use an optimizer such as Muon and that Muon is bringing a better optimization trajectory in some way by making sure stable rank does not degrade. A similar perspective, through the viewpoint of curvature, exists in the paper I mention above  "Preconditioners for the Stochastic Training of Neural Fields" in CVPR 2025 by Chng et al.

2. **Soundness:** Another weakness of the paper is that the statements are not really backed up with any theoretical arguments. While this is not always an issue and empirical work is fine. The issue is that their methodology is to just apply Muon which is an already known optimizer. Thus their statements on spectral bias would have made more of an impact had they obtained some theoretical arguments to back up what they are saying.

---

> ### Author Rebuttal · Authors · 2026-03-31
>
> We appreciate your feedback and provide point-by-point answers to your questions below.
>
> >Significance: The major issue I feel with this paper is the significance of the contribution.
>
> We thank you for the feedback, but would like to clarify the significance of our contribution, which bridges a fundamental theoretical gap with substantial empirical results:
>
> First, it provides a unifying theory based on stable rank that simultaneously *explains improvements across input-, activation-, and optimization-level interventions*.
>
> Second, and most crucially for practitioners, this framework directly enables up to a +9 dB improvement in PSNR for standard architectures like FFNs. This is one of the largest empirical leaps reported in the INR literature to date, highlighting that our work offers substantial, immediate value to both theoretical and applied INR research.
>
> > What is the real contribution of your work? [...]
>
> Our core contribution is to identify stable rank degradation as a mechanistic and actionable driver of spectral bias.
> This shifts the focus from architecture design to training dynamics, providing a) diagnostic tool (monitoring during training), b) a unifying explanation across methods, and c) a new direction for optimization and regularization in INRs.
>
> > [...] statements are not really backed up with any theoretical arguments.
>
> We respectfully disagree with this assessment. Our work provides concrete theoretical grounding for the mechanism of spectral bias and stable rank.
>
> Equation (1) shows that the row space of each layer's gradient update is contained in the column space of its activations, Equation (2) formalizes the resulting upper bound on the stable rank of updates.
>
> For INRs, the analysis in Section 3.1 shows how this constraint propagates across layers, leading to progressively restricted update directions during training. Importantly, this analysis is independent of any specific optimizer or regularization method. This is exactly why we start with pretraining [1] and normalization [2] in Section 3.3, and select the optimizer as the final of three rank-preserving interventions we study.
>
> > I found the work [...] by Chng et al.
>
> Thank you for sharing this paper. Chng et al. [3] offer a complementary perspective linking activation choice to Hessian conditioning, but focus on training speed rather than providing a unifying framework that spans all three axis of INR design. We will add this to our discussion.
>
> > [...] you don't really have any limitations of the actual method.[...]  Newton-Schulz [...] make the method very efficient?
>
> Key limitations are that (i) optimal stable rank remains unknown and that (ii) optimization-level interventions narrow, but do not fully close gaps (see RegR). Regrading compute overhead, Newton-Schulz is used as an efficiency trick with little runtime overhead [4]. We will discuss NS approximation and limitations in more detail in the updated manuscript.
>
> > [...] this already shows that Adam is doing some sort of gradient regularization so does this not mean Muon is just another level of this?
>
> You are correct that orthogonalization may be viewed as another form of gradient regularization.
>
> However, and importantly, our analysis **goes beyond Muon**, which is why we use stable rank to explain (i) **input-level** and (ii) **activation-level** interventions using this holistic perspective. We further examine **pretraining [1]** and **normalization [2]** as instances of (iii) optimization-level interventions, which go beyond optimizers and their gradient landscapes.
>
> > It would have been good had you put SGD in the experiments too but I believe on tasks like NeRF it is impossible to train an INR with SGD.
>
> We add an optimizer ablation (Adam, Lion, and Muon) in the reply to reviewer F9g6.  Lion, while competitive to Adam, does not increase stable rank and falls short of Muon in rank preservation and reconstruction. Thus, it is beneficial to use methods that increase stable rank *by design*.
>
> > [...] Doesn't this mean that while Muon is helping ReLU MLP overcome spectral bias it is not doing it as efficiently as say SIREN?
>
> This is *exactly* an insight the reader should take away! Optimization is an orthogonal, underappreciated perspective we aim to bring to the INR community for discussion. Hence, it does not replace other interventions; rather, it complements them. As we show, combining interventions yields best results.
>
> We thank you for feedback and are happy to discuss further aspects during the rebuttal.
>
> ---
> [1] Daneshmand, Hadi, et al. "Batch normalization provably avoids ranks collapse for randomly initialised deep networks." NeurIPS 2020
>
> [2] Cai, Zhicheng, et al. Batch normalization alleviates the spectral bias in coordinate networks. In: CVPR 2024
>
> [3] Chng, Shin-Fang et al.. "Preconditioners for the stochastic training of neural fields." CVPR 2025.
>
> [4] Jeremy Bernstein, “Deriving Muon”, URL: https://jeremybernste.in/writing/deriving-muon,

---

> > ### Author Rebuttal · Reviewer_zca9 · 2026-04-02
> >
> > I thank the authors for their rebuttal. It's my belief that this paper is not at the level of ICML. While the paper has some good contributions I cannot say that the theoretical contributions is one of them and would rather say that the empirical results are what is the standout for the paper. However, the authors have not done enough of a job of explaining why those empirical results come about. While they have equations (1) and (2) these are simple equations that cannot be claimed as a contribution. This is also not in line with the papers the authors cite, for example I looked up the paper "Curvature-aware training for coordinate networks" that the authors cite as one of the papers that speaks about optimization for INRs (aka coordinate networks) and it's theoretical contributions have explicit theorems and proofs that explain how an optimizer helps particular class of INRs. Other papers that the authors cite such as "Wavelet implicit neural representations" construct something completely new, in this case a new activation, and are thus bringing a new construction to the community. The authors paper does none of this. They simply provide a simple insight and suggest that is a key driver of spectral bias which can be then maximised through an optimizer and then suggest that there are already optimizers doing that such as Muon so people should use optimizers such as Muon. There is no real core contribution.
> > In order for this paper to be accepted into a top conference venue such as ICML it is my viewpoint that the paper needs a higher contribution. Therefore, I will keep my score.
> >
> > **We thank you for the feedback, but would like to clarify the significance of our contribution, which bridges a fundamental theoretical gap with substantial empirical results:...**
> >
> > I completely disagree with the authors here. First of all I don't agree that there is a fundamental theoretical gap in the literature and from my knowledge of the literature the authors are over playing their contributions. Furthermore, they do not bridge a theoretical gap as the theoretical contribution of this work is minor. Their theoretical contribution is given by equation (1) which is a simple gradient computation and equation (2) provides an upper bound on the stable rank of the updates and this uses simple properties of rank. There is nothing deep here. Suggesting that somehow this solves a major theoretical gap in the literature about INRs is far overplaying the contributions.
> >
> > **Our core contribution is to identify stable rank degradation as a mechanistic and actionable driver of spectral bias. This shifts the focus from architecture design to training dynamics, providing a) diagnostic tool (monitoring during training), b) a unifying explanation across methods, and c) a new direction for optimization and regularization in INRs.**
> >
> > However, you don't actually say anything about the actual training dynamics. You don't actually theoretically show this. You compute an upper bound that suggests this but the theory falls short in actually showing it.
> >
> > **Key limitations are that (i) optimal stable rank remains unknown and that (ii) optimization-level interventions narrow, but do not fully close gaps (see RegR). Regrading compute overhead, Newton-Schulz is used as an efficiency trick with little runtime overhead [4]. We will discuss NS approximation and limitations in more detail in the updated manuscript.**
> >
> > Thank you for pointing out these limitations.
> >
> > **However, and importantly, our analysis goes beyond Muon, which is why we use stable rank to explain (i) input-level and (ii) activation-level interventions using this holistic perspective. We further examine pretraining [1] and normalization [2] as instances of (iii) optimization-level interventions, which go beyond optimizers and their gradient landscapes.**
> >
> > I don't see you analysis going beyond Muon. What your analysis is trying to do is show that stable rank plays a role in spectral bias and if we can find an optimal stable rank (which you do not even do) then one can overcome spectral bias through the optimizer by forcing isotropy within the gradient updates. Optimizers like Muon already do this all you are doing is saying that this is then good for stable rank, which comes down to equation (1) and (2).
> >
> > **This is exactly an insight the reader should take away! Optimization is an orthogonal, underappreciated perspective we aim to bring to the INR community for discussion. Hence, it does not replace other interventions; rather, it complements them. As we show, combining interventions yields best results.**
> >
> > Thank you for clearing this up. When I initially read your paper I got the perspective that you were trying to say that the optimizer can replace the non-standard activations like SIREN but I am glad you cleared that up.

---

> > > ### Author Response · Authors · 2026-04-03
> > >
> > > Thank you for the continued discussion and for engaging with our rebuttal. We are glad our rebuttal clarified that our core contribution is a framework based on three complementary angles of INR design, i.e., input-level, activation-level, and importantly, optimization-level interventions, using a stable rank perspective to explain and investigate the spectral bias. This is not a minor finding: As you noted in your initial review, studying and mitigating spectral bias in INRs via optimizer design is novel in this context, and this contribution was recognized as adding value to the INR community. The other reviewers independently echo this assessment, describing the paper as offering a "new perspective" (RgeR, F9g6) with potential for "high impact on future research" (Cui8).
> > >
> > > We would also like to emphasize that this paper was submitted to the **Applications->Computer Vision** area, where practical impact is a primary criterion. On that dimension, our contribution is strong: across multiple use cases and established benchmarks, our method yields improvements of up to 9 dB PSNR, which are, to our knowledge, among the strongest reported in the INR literature. These are not marginal gains, but substantial improvements that demonstrate immediate practical value for INR-based vision applications.
> > >
> > > We agree that a theoretical analysis is interesting. However, for multilayer INRs, relating activations to weight (stable) rank requires strong assumptions (e.g., well-conditioned inputs and non-saturating activations), which are hard to justify after training. Even then, the resulting bounds are typically loose and expressed via proxy quantities, limiting their explanatory value. As such, we believe adding this analysis would not meaningfully strengthen our empirical broadly applicable contribution.
> > >
> > > For these reasons, we strongly believe the paper makes a _clear contribution for this track_: it introduces a novel and useful perspective on INR design, identifies optimization-level intervention as an effective and underexplored mechanism for addressing spectral bias, and supports these insights with unusually strong empirical evidence.

---

### Official Review · Reviewer_F9g6 · 2026-03-11

**Soundness:** 2
**Presentation:** 3
**Significance:** 3
**Originality:** 3
**Overall Recommendation:** 3
**Confidence:** 4

**Summary:**

This paper studies the problem of Implicit Neural Representations and challenges the convention that MLP-based INRs are incapable of representing high-frequency content. The authors find that regulating the network's rank during training substantially improves the fidelity of the learned signal, rendering even simple MLP architectures expressive. Therefore, the authors propose to replace Adam optimizer with Muon during training. Empirical results on multiple datasets and tasks demonstrate the effectiveness and generalization ability of the proposed method.

**Compliance With Llm Reviewing Policy:**

Affirmed.

**Final Justification:**

I thank the authors for the response. However, my concern remains unresolved. The NeRF experiments still do not convincingly demonstrate improvement over the official FINER baseline, as the reproduced Adam baseline underperforms the published FINER results on most scenes, and Muon does not consistently exceed them. Consequently, the empirical evidence mainly shows gains over the authors’ own reruns rather than over the prior state-of-the-art. Additionally, the sensitivity of the reproduced baseline to the training setup makes the source of the observed gains difficult to attribute.

Therefore, I maintain my original rating.

**Key Questions For Authors:**

Please refer to weakness 1,2,3.

**Limitations:**

Yes

**Strengths And Weaknesses:**

Strengths:
1. The authors provide a new perspective on the low-frequency bias of vanilla MLPs and find that the problem is caused by a symptom of stable rank degradation during training. This may provide insights into future work in this area.
2. The experiments are extensive. The authors have conducted comprehensive experiments across different tasks, datasets, and model architectures to demonstrate the effectiveness and generalization ability of the method.
3. The presentation seems good. Overall, the paper is well-organized and the writing is clear.

Weaknesses:
1. The technical contribution seems limited. While the finding has its significance, the only solution provided in the paper is using Muon as the optimizer during training. Although it has shown improvement, it remains unverified whether it is the optimal solution to the problem.
2. A follow-up question is that, it seems that there is no ablation on the choices of optimizers, apart from Adam and Muon. Given that the optimizer affects the result significantly, whether other optimizers like Lion, Sophia, and AdamW would alleviate the issue remains unverified.
3. Moreover, different optimizers may need different training strategies or setups to ensure optimal performance, and it is unclear whether the authors actually find the optimal setup. A reference could be, replicating the experimental setup in recent state-of-the-art INR methods and observing if changing the optimizer to Muon could lead to further improvement.

---

> ### Author Rebuttal · Authors · 2026-03-31
>
> We thank you for the positive assessment and recognizing the paper's *"new perspective"* and *“extensive experiments”*.
>
> > [...] no ablation on the choices of optimizers, apart from Adam and Muon. Given that the optimizer affects the result significantly, whether other optimizers like Lion, Sophia, and AdamW would alleviate the issue remains unverified.
>
> While our framework broadly addresses rank-preserving mechanisms, and Section 3 evaluates normalization [1] and pretraining [2] (both consistently less effective than optimizer interventions, Figure 2), we focus our experiments on optimizers as the most direct lever.
>
> Following your suggestion, we added an additional optimizer. Rather than an exhaustive benchmark, we aim to isolate rank-preserving updates. AdamW shares Adam’s update subspace and *does not test a new hypothesis*, while Sophia's search space requires extensive tuning for fair comparison. We thus focus on qualitatively distinct mechanisms:  adaptive (Adam), sign-based (Lion), and orthogonalized (Muon), to test our central claim.
>
> **Table 1**: Stable rank of hidden layers at convergence, averaged over 24 Kodak images.
> |Model|L1 Adam|L1 Lion|L1 Muon|L2 Adam|L2 Lion|L2 Muon|
> |---|---|---|---|---|---|---|
> |ReLU MLP|4.31±1.56|1.46±0.11|**18.79±3.29**|8.43±1.49|2.13±0.12|**39.18±6.26**|
> |ReLU FFN|5.86±0.54|1.06±0.03|**34.13±14.51**|6.28±1.76|1.61±0.07|**15.90±4.73**|
> |Gaussian MLP|42.94±9.16|8.53±0.97|**64.14±10.42**|17.27±3.45|1.50±0.19|**58.90±7.33**|
> |Gaussian FFN|**77.95±0.35**|22.48±25.28|74.92±4.29|**61.33±3.36**|1.08±0.02|49.64±12.95|
> |SIREN|16.00±5.76|26.35±5.61|**44.67±6.72**|1.88±0.26|9.52±3.79|**60.69±9.22**|
> |WIRE|36.70±10.94|6.84±0.87|**50.79±5.43**|29.51±17.04|59.83±8.53|**64.50±5.34**|
> |FINER|63.33±2.68|35.88±10.37|**73.35±4.34**|7.41±0.70|13.70±12.27|**70.32±6.59**|
>
> **Table 2**: Image overfitting performance (PSNR ↑).
> |Model|Adam|Lion|Muon|
> |---|---|---|---|
> |ReLU MLP|22.83±2.85|24.34±3.08|**30.04±3.59**|
> |ReLU FFN|31.21±2.71|30.63±3.42|**40.50±3.29**|
> |Gaussian MLP|35.28±1.85|35.25±2.73|**41.05±3.06**|
> |Gaussian FFN|25.20±2.18|24.69±2.12|**28.13±2.08**|
> |SIREN|39.33±2.64|41.75±2.96|**42.33±3.29**|
> |WIRE|32.43±9.56|39.73±3.36|**42.99±3.34**|
> |FINER|39.87±2.04|42.46±3.40|**42.59±2.74**|
>
> Two consistent patterns emerge:
>
> (1) **Optimization controls rank independently of architecture**. Lion does not consistently preserve stable rank across layers and underperforms Muon in both rank and reconstruction quality. For vanilla ReLU MLPs, Muon mitigates rank collapse and improves PSNR from 22.8 to 30.0 dB.
>
> (2) Input- and activation-level interventions **reshape where rank is expressed** but **do not prevent collapse**. FFN and SIREN/WIRE/FINER models concentrate high rank in early layers but degrade deeper. Muon preserves high rank throughout; Lion provides only partial improvement.
>
> Thus, a single optimizer change yields **up to +9 dB PSNR** without architectural changes or additional parameters.
>
> > [...] different optimizers may need different training strategies or setups to ensure optimal performance [...]
>
> As detailed in the original manuscript, we ensured optimal training setups for all tested optimizers and rank-preserving methods, using established strategies from recent benchmark papers [3,4] (Sections A, B) and conducted exhaustive hyperparameter sweeps for each optimizer, architecture, and task, as documented in e.g. Table 7. We hope this addresses the reviewer's concerns.
>
> > [...] technical contribution seems limited. While the finding has its significance, the only solution provided in the paper is using Muon as the optimizer during training. [...]
>
> Previous research has focused on two axes to improve INR expressiveness, namely at (i) **input level** (e.g., FFNs) and (ii) **activation level** (e.g., SIRENs). Here, we identify stable rank collapse as the hidden bottleneck to INR expressiveness. While we find Muon superior to pretraining and normalization [1,2], our core contribution is revealing rank-preserving optimization as a **third, previously overlooked axis** for scaling model capacity. Importantly, we show that this **novel axis** is additive to the established axes and broadly applicable.
>
> We believe the newly added ablations strengthen the paper and are happy to discuss further.
>
> ---
>
> [1] Daneshmand, Hadi, et al. "Batch normalization provably avoids ranks collapse for randomly initialised deep networks." Advances in Neural Information Processing Systems 33 (2020): 18387-18398.
>
> [2]  Cai, Zhicheng, et al. Batch normalization alleviates the spectral bias in coordinate networks. In: Proceedings of the IEEE/CVF conference on computer vision and pattern recognition. 2024. S. 25160-25171.
>
> [3] Kim, Namhoon, and Sara Fridovich-Keil. "Grids Often Outperform Implicit Neural Representation at Compressing Dense Signals." NeurIPS 2025
>
> [4] Essakine, Amer, et al. "Where do we stand with implicit neural representations? a technical and performance survey." TMLR 2024

---

> > ### Author Rebuttal · Reviewer_F9g6 · 2026-04-04
> >
> > Thanks for the detailed responses from the authors. While the additional experiments addressed my concerns about optimizer variances, I still hold concerns about the significance of this problem. If the problem is a fundamental and overlooked issue in INRs, the proposed method should generalize to all existing methods. That said, the proposed approach could be applied to recent state-of-the-art methods and achieve even better performance. However, this part still remains unverified, so I tend to keep my rating.

---

> > > ### Author Response · Authors · 2026-04-04
> > >
> > > Thank you for your response. We are pleased that our additional experiments successfully addressed your concerns regarding optimizer variance.
> > >
> > > We would like to emphasize that the generalizability of our framework has already been demonstrated in the paper. To ensure both broad relevance and reproducibility, we deliberately base our evaluation on _recent large-scale INR benchmarks [1,2]_, which reflect current state-of-the-art practices across tasks such as novel view synthesis, CT reconstruction, super-resolution, and image/audio representation. Importantly, we closely follow these reference setups to ensure a fair comparison and _avoid introducing evaluation bias_, exactly as you suggested in your initial review.
> > >
> > > Our experiments, therefore, do not rely on simplified settings, but incorporate recognized state-of-the-art methods, including:
> > > - **State-of-the-art Input-level interventions**: positional encoding, Fourier features, Instant-NGP
> > > - **State-of-the-art Activation-level interventions**: SIREN, WIRE, FINER, Gaussian MLPs
> > >
> > > Across these recent methods, we consistently observe that rank-preserving optimization improves both stable rank and reconstruction quality (e.g., up to +9 dB PSNR), without requiring architectural changes. Crucially, these optimization-level interventions do not replace either input-level or activation-level interventions, but are synergistic with them - showing, again, the _broad applicability and generalizability_ of our framework.
> > >
> > > We hope this resolves your remaining concerns, and we are happy to engage further if any additional clarifications are needed.
> > >
> > > [1] Kim, Namhoon, and Sara Fridovich-Keil. "Grids Often Outperform Implicit Neural Representation at Compressing Dense Signals." NeurIPS 2025
> > >
> > > [2] Essakine, Amer, et al. "Where do we stand with implicit neural representations? a technical and performance survey." TMLR 2025

---

### Official Review · Reviewer_RgeR · 2026-03-13

**Soundness:** 3
**Presentation:** 3
**Significance:** 2
**Originality:** 2
**Overall Recommendation:** 4
**Confidence:** 4

**Summary:**

This paper challenges the widespread notion of low-frequency bias of MLP-based INRs being an architectural limitation, and rather frames it as a symptom of stable rank degradation during training. Authors provide theoretical grounding on this phenomenon, discussing different interventions to increase stable rank. Finally, they outline an optimizer-level intervention to achieve better representation quality.

**Compliance With Llm Reviewing Policy:**

Affirmed.

**Final Justification:**

I increase my score following the rebuttal period and additional experiments.

**Key Questions For Authors:**

How would this analysis carry over to plane-based formulations such as K-Planes, Tri-Planes, TensoRF, GA-Planes? Can you provide mathematical analysis/intuition based on the selected resolution/channel dimensions, choice of feature combination and MLP decoder hidden dimension? Do these models still suffer from stable rank decay? Can you provide some experimental comparisons (as you have done for Fourier encoding)?

**Limitations:**

The authors should add a limitations discussion.

**Strengths And Weaknesses:**

Soundness: The paper is technically sound. There is mathematical grounding as well as sufficient experimentation.

Presentation: The paper is easy to read and follow. The work discusses previous approaches and how it brings a new perspective.

Significance: The paper does address an important problem and advances understanding regarding the spectral bias limitations of INR models.

Originality: The main novelty of the paper is the way they explain stable rank decay during training. Both INRs and the discussed optimizer Muon have been established. The presented perspective is new, however the use of Muon optimizer with different INR variants feels a bit incremental.

---

> ### Author Rebuttal · Authors · 2026-03-31
>
> Thank you for your thoughtful feedback and for recognizing that our paper “advances understanding regarding the spectral bias”.
>
> > How would this analysis carry over to plane-based formulations [...]? Do these [...] suffer from stable rank decay?
>
> Plane-based methods act as input-level interventions (Sec. 3.2), injecting high-rank structure via structured grids before the INR's MLP decoder. While high-rank input structure increases input rank (F), it does not prevent rank collapse in the decoder itself during optimization. So, the short answer is: **Yes, they still suffer from stable rank decay.**
>
> > Can you provide experimental comparisons?
>
> We evaluate GA-Planes (which generalizes K-/Tri-Planes) under the same setup as Sec. 4.1.1. While structured inputs alone do not prevent rank collapse, rank-preserving optimization consistently improves both stable rank and reconstruction quality.
>
> |Method|Feat. Comb.|Opt.|Stable Rank ↑|PSNR ↑|
> |---|---|---|---|---|
> |ReLU MLP|-|Adam|3.82±1.39|22.83±2.85|
> |ReLU MLP|-|Muon|28.29±3.04|30.04±3.59|
> |GA-Planes|Sum|Adam|5.75±0.66|33.66±2.55|
> |GA-Planes|Sum|Muon|**27.24±3.17**|**35.70±3.24**|
> |GA-Planes|Multiply|Adam|5.24±0.67|37.19±3.00|
> |GA-Planes|Multiply|Muon|**20.16±3.39**|**38.57±3.26**|
>
> **Key results**:
> - Under Adam, stable rank remains very low ( approx. 5-6) despite structured inputs.
> - Orthogonalized optimization increases stable rank by approx. 4-fold (20-25 range)
> - Importantly, combining structured inputs and orthogonalized optimization achieves the highest downstream metrics.
>
> These results *validate* three predictions of our framework:
> - Feature combination sets the rank ceiling
> - Summation → additive constraint (low ceiling)
> - Hadamard product → multiplicative capacity (higher ceiling)
>  → Explains +3.53 dB gap (Multiply vs Sum under Adam)
>
> Optimization matters more *under tighter bottlenecks*:
> - Muon gains are larger for summation (+2.04 dB, 4.7x rank)
> - Low ceilings leave more room for recovery
> - Both axes are necessary
> - **Best**: Multiply+Muon (38.57 dB)
>
> **Takeaway:** Input design and optimization are complementary, and both are required.
>
>
> > Can you provide intuition based on the [...] MLP decoder hidden dimension?
>
> |Hidden Dim $h$|Stable Rank ↑|PSNR ↑|
> |---|---|---|
> |32|7.23|24.98|
> |64|14.23|27.44|
> |128|17.42|30.66|
>
> Increasing h consistently improves stable rank and quality, confirming decoder width is a hard bottleneck as predicted by our bound.
>
> > [...] provide mathematical analysis?
>
> $\nabla_{W_l} \mathcal{L} = H_{l-1}^T G_l \Rightarrow \text{rank}(\nabla_{W_l} \mathcal{L}) \leq \min(\text{rank}(F), h).$
>
> Decoder width $h$ bounds maximum rank; feature combination controls rank growth and high-rank inputs can still collapse during optimization.
>
> > The use of Muon optimizer [...] feels a bit incremental.
>
> We respectfully clarify that our contribution is not the use of a specific optimizer, but identifying a previously overlooked axis in the INR design space. Prior work focuses on (1) input-level interventions (Fourier, Hash Grids, planes) and (2) activation-level interventions (SIREN, WIRE, FINER). We introduce stable rank optimization as a third, orthogonal axis, and show that stable rank collapse is a key bottleneck to INR expressiveness. We use Muon because it best prevents this collapse among rank-preserving strategies (pretraining [1], batch normalization [2], Sec. 3.1).
>
> > [...] add a limitations discussion.
>
> Thank you for pointing this out. We will add limitations, with the key points being:
>
> (i) Rank decay mitigation improves high-frequency reconstruction, but optimal stable rank remains unknown.
>
> (ii) Optimization-level interventions narrow, but do not fully close gaps, to more expressive architectures.
>
> **Additional Experiments:**
>
> Additional experiments confirm framework generalizes across architectures, inputs, and optimizers:
>
> (i) Optimizer ablation (Adam, Lion, Muon) across 7 architectures on Kodak (Tables 1, 2 in response to F9g6). Results confirm performance correlates with optimizer's ability to maintain a high stable rank, particularly for architectures without other interventions.
>
> (ii) Embedding hyperparameter sweep (response to Cui8). Orthogonalized optimization outperforms Adam across all tested sigma values, confirming optimizer-level improvements are additive.
>
> We believe these results, together with the GA-Planes ablation above, significantly strengthen the paper and demonstrate that the framework generalizes across architectures, input representations, and optimizers.
>
> We hope our rebuttal addresses your concerns and welcome further discussion.
>
> ---
>
> [1] Daneshmand, Hadi, et al. "Batch normalization provably avoids ranks collapse for randomly initialised deep networks." NeurIPS 2020
>
> [2] Cai, Zhicheng, et al. Batch normalization alleviates the spectral bias in coordinate networks. CVPR’24

---

> > ### Author Rebuttal · Reviewer_RgeR · 2026-04-04
> >
> > I thank the reviewers for their experimentally grounded response. I increase my score by 1.

---

> > > ### Author Response · Authors · 2026-04-04
> > >
> > > Thank you very much for your valuable feedback throughout the rebuttal and discussion phase. We are happy to see that all concerns have been addressed. We will include the rebuttal, including your proposed ablation on GA-Planes, in the revised version.

---

### Decision · Program_Chairs · 2026-04-30

**Decision:**

Accept (regular)

**Comment:**

The submission received a split set of scores: Weak Accept (RgeR), Weak Reject (F9g6), Reject (zca9), and Accept (Cui8), all with high confidence. Most reviewers appreciate the novel perspective of addressing low-frequency bias by regulating the network’s rank, as well as the significant improvements over baselines across various tasks. However, they also raised concerns regarding:

- the use of existing optimizers, which limits perceived technical contribution;

- the theoretical analysis, which is not entirely convincing.

In the rebuttal, the authors incorporated hyperparameter searches to demonstrate that experimental parameters were not cherry-picked.
While reviewers generally agree on the paper’s merits, they assign significantly different weights to its insight and technical contribution. The Area Chair finds that the submission provides valuable insight and convincing results. The finding that stable rank plays a role in spectral bias may inspire future research, and thus the AC recommends acceptance at this stage.

The authors are required to integrate all new experiments and clarifications from the rebuttal into the final manuscript. Please ensure the material is clearly organized and presented in the camera-ready version.